REPORT

# The dynamics of centromere assembly and disassembly during quiescence

Océane Marescal[1,2], Kuan-Chung Su[1], Brittania Moodie[1], Noah J.L. Taylor[1,2], and Iain M. Cheeseman[1,2]

**Quiescence is a state in which cells undergo a proliferative arrest while maintaining their capacity to divide again. Here, we analyze how cells regulate their centromeres during quiescence entry and exit. Despite the constitutive localization of centromere proteins in proliferating cells, cells rapidly disassemble most centromere proteins during quiescence entry while preserving those required to maintain centromere identity. We show that this disassembly occurs primarily through the transcriptional downregulation of centromere proteins. During quiescence exit, the centromere is reassembled during the first S phase to regain normal homeostatic centromere protein levels. CENP-A is typically deposited during G1. However, we find that CENP-A deposition does not occur during the G1 immediately following quiescence exit and instead occurs in the G1 after cells complete their first mitosis. We find that the presence of PLK1 distinguishes these distinct G1 states. These findings reveal centromere dynamics during quiescence entry and exit and highlight paradigms for controlling centromere assembly and disassembly.**

## Introduction

Quiescence is a state of reversible proliferative arrest in which cells are no longer dividing but retain the capacity to reenter the cell cycle (Marescal and Cheeseman, 2020). Examples of quiescent cells include hepatocytes, oocytes, and tissue-resident stem cells, all of which can remain in a nondividing state for years, only reentering the cell cycle when activated with the appropriate stimulus (Berasain and Avila, 2015; de Morree and Rando, 2023; Kang et al., 2025; Kim and You, 2022; Lei et al., 2024; Marescal and Cheeseman, 2020; Peng et al., 2024; Schwabe and Brenner, 2025; Urban et al., 2019; van Velthoven and Rando, 2019; Zhao et al., 2024). Thus, quiescent cells are faced with the unique task of preventing cell division while simultaneously preserving their capacity to divide. This need to achieve a balance between growth arrest and the capacity for cell cycle reentry is reflected in the numerous changes to gene expression and metabolism that characterize the quiescent state (Chapman et al., 2020; Cheung and Rando, 2013; Coller et al., 2006; Du et al., 2023; Fukada et al., 2007; Ho et al., 2017; Johnson et al., 2017; Kang et al., 2024; Liang et al., 2020; Marescal and Cheeseman, 2020; Min and Spencer, 2019; Mitra et al., 2025; Subramaniam et al., 2014). Quiescent cells also undergo alterations to cellular structures and organelles, including centrioles, nuclear pores, and mitochondria (Azizzanjani et al., 2025; Baker et al., 2022; Breslow and Holland, 2019; Coller, 2019; Feldherr and Akin, 1991; Feldherr and Akin, 1993; Maeshima et al., 2006; Palla et al., 2022; Venugopal et al., 2020; Wang et al., 2024).

One other key cellular structure is the centromere. During mitosis, the centromere recruits the macromolecular kinetochore complex, which in turn is crucial for mediating microtubule attachments and segregating the genetic material (McKinley and Cheeseman, 2016). In vertebrate cells, centromeres are specified epigenetically through the centromere-specific histone H3 variant, CENP-A (Barnhart et al., 2011; McKinley and Cheeseman, 2016; Mendiburo et al., 2011; Palmer et al., 1991; Stirpe and Heun, 2023; Vafa and Sullivan, 1997). Previous work from our lab and others found that quiescent cells slowly but continuously incorporate new CENP-A nucleosomes at centromeric regions to maintain centromere identity during quiescent arrest (Saayman et al., 2023; Swartz et al., 2019). In addition to CENP-A, 16 other CENP proteins localize constitutively to the centromere in all phases of the cell cycle in proliferating cells (Cheeseman and Desai, 2008). These proteins, collectively known as the constitutive centromere-associated network (CCAN), can be divided into five groups that are interdependent on each other for localization: CENP-C, the CENP–L/N complex, the CENP–H/I/K/M complex, the CENP–O/P/Q/U/R complex, and the CENP–T/W/S/X complex (Cheeseman and Desai, 2008; McKinley and Cheeseman, 2016). CCAN components recruit microtubule-binding proteins to the

[1]Whitehead Institute for Biomedical Research, Cambridge, MA, USA;   [2]Department of Biology, Massachusetts Institute of Technology, Cambridge, MA, USA.

Correspondence to Iain M. Cheeseman: icheese@wi.mit.edu

I.M. Cheeseman is the lead contact.

chromosomes during mitosis, forming the centromere-kinetochore interface (Hori et al., 2008; McKinley et al., 2015; Nishino et al., 2012; Przewloka et al., 2011; Screpanti et al., 2011; Sissoko et al., 2024). The fate of these CCAN components as cells enter and exit quiescence remains poorly understood.

Here, we investigate the behavior and regulation of centromere components during quiescence entry and exit. Despite their constitutive localization in cycling cells, we find that centromere proteins are rapidly lost upon quiescence entry. This centromere disassembly is regulated at the level of transcription through the downregulation of most CCAN components, such that the ectopic expression of selected CCAN proteins can restore their localization. In contrast, we show that CENP-C is uniquely preserved at low levels in quiescent cells where it contributes to CENP-A deposition. Finally, we analyze the dynamics of centromere reassembly during cell cycle reentry and exit from quiescence. We find that most CCAN proteins are redeposited at centromeres during the S phase immediately following quiescence exit in a process that occurs independently of DNA replication. Conversely, new CENP-A is not deposited until after the first mitotic division following quiescence release, with prior deposition restricted in part by the absence of Plk1 kinase. Together, this study defines centromere protein behavior and dynamics during quiescence and highlights how quiescent cells can modify a cellular structure to balance growth arrest with the capacity to reenter the cell cycle.

## Results and discussion

### The centromere is rapidly disassembled upon quiescence entry

To analyze the behavior of centromere proteins in quiescent cells, we induced non-transformed human retinal pigment epithelial (RPE1) cells to enter quiescence using a combination of serum starvation and contact inhibition. This treatment successfully induced quiescence, as monitored by lack of 5-ethynyl-2′-deoxyuridine (EdU) incorporation and 2n DNA content (Fig. S1, A–C). Following quiescence induction, we used immunofluorescence to monitor the localization of CCAN proteins over the course of 7 days. The localization of the centromeric histone H3 variant, CENP-A, persisted during quiescence (Fig. S1 D), as described previously (Swartz et al., 2019). In contrast, the rest of the centromere CCAN proteins were rapidly disassembled upon entry into quiescence (Fig. 1, A–J). In particular, CENP-T, CENP-L, and CENP-K showed a marked reduction of centromere intensity within just 24 h of quiescence induction and were completely lost after 5 days (Fig. 1, A–F). CENP-O/P levels remained stable for a longer period but were eventually lost by 5 days of quiescence (Fig. 1, G and H). Although CENP-C centromere intensity also decreased in quiescent cells, its levels plateaued to around one-fourth of that found in cycling cells, maintaining diminished but stable centromere localization during quiescence (Fig. 1, I and J). This pattern of quiescence CCAN behavior was conserved in mouse 3T3 cells (Fig. S1, E–H), which also showed CENP-T loss following quiescence entry, but CENP-C retention.

In addition to measuring centromere localization, we conducted western blotting to monitor centromere protein levels (Fig. 1, K and L; and Fig. S1, I–K). CENP-T and CENP-H proteins were depleted in quiescent cells, whereas CENP-C protein was retained, consistent with their localization behavior. Thus, centromere components are rapidly disassembled upon quiescence entry through the depletion of most centromere components, whereas CENP-C is retained with reduced centromere levels.

### Quiescent cells regulate centromere behavior through a transcriptional program

To determine how centromere disassembly is regulated, we next conducted RNA sequencing in quiescent and cycling cells (Fig. 2 A and Fig. S2 A). We observed a strong decrease in the expression of most centromere protein mRNAs in quiescent cells, with the exception of CENPC mRNA, which was increased by around twofold. We additionally verified these results using qPCR (Fig. 2, A–C). We also observed similar trends in mouse 3T3 cells (Fig. S2 B). These changes in mRNA levels occurred within 24 h of quiescence induction, closely mirroring the dynamics of the loss of CCAN protein levels and localization (Fig. 2 C).

Decreases in centromere mRNA levels could be due to downregulated transcription or to reduced RNA stability. To test whether CENP mRNAs were less stable in quiescent cells, we treated cycling and quiescent cells with the transcriptional inhibitor actinomycin D and measured the change in mRNA levels over time (Fig. 2, D–H; and Fig. S2 C [Dani et al., 1984]). We found that centromere mRNAs showed similar rates of degradation in both cycling and quiescent cells. Thus, decreased centromere mRNA levels in quiescence are not due to decreased RNA stability but instead reflect decreased transcription.

We next considered whether the loss of CCAN protein localization is driven by this reduction in gene expression. The CENP–L/N complex is completely lost from centromeres and shows drastically decreased mRNA levels in quiescent cells (Fig. 1, C and D; and Fig. 2, A–C). To test if CENP–L/N delocalization is due to reduced gene expression during quiescence, we ectopically expressed GFP-CENP-N in RPE1 cells. Of all the CCAN proteins, we chose to ectopically express CENP-N, as CENP-N binds directly to CENP-A nucleosomes (Carroll et al., 2009; Carroll et al., 2010) and requires CENP-C and CENP-A for its recruitment to centromeres (McKinley et al., 2015). As quiescent cells retain both CENP-C and CENP-A (Fig. 1, I and J; and Fig. S1 D), ectopic CENP-N expression would be predicted to be able to localize to centromeres during quiescence. Indeed, we found that quiescent cells expressing GFP-CENP-N retained CENP-N centromere localization (Fig. 2, I and J). This localization was slightly reduced when compared with cycling cells, which could be due to the lack of the CENP-N–binding partner, CENP-L, in quiescent cells or to reduced levels of CENP-C. Thus, centromere proteins are capable of being recruited to centromeres during quiescence but fail to do so because of their downregulated gene expression. Together, these data suggest that the regulation of centromere disassembly during quiescence entry likely occurs, at least in part, through a rapidly regulated transcriptional program.

### FoxM1 transcription factor may regulate centromere protein gene expression during quiescence

Given the coordinated gene expression behavior for most CCAN genes, we next considered whether specific pathways or

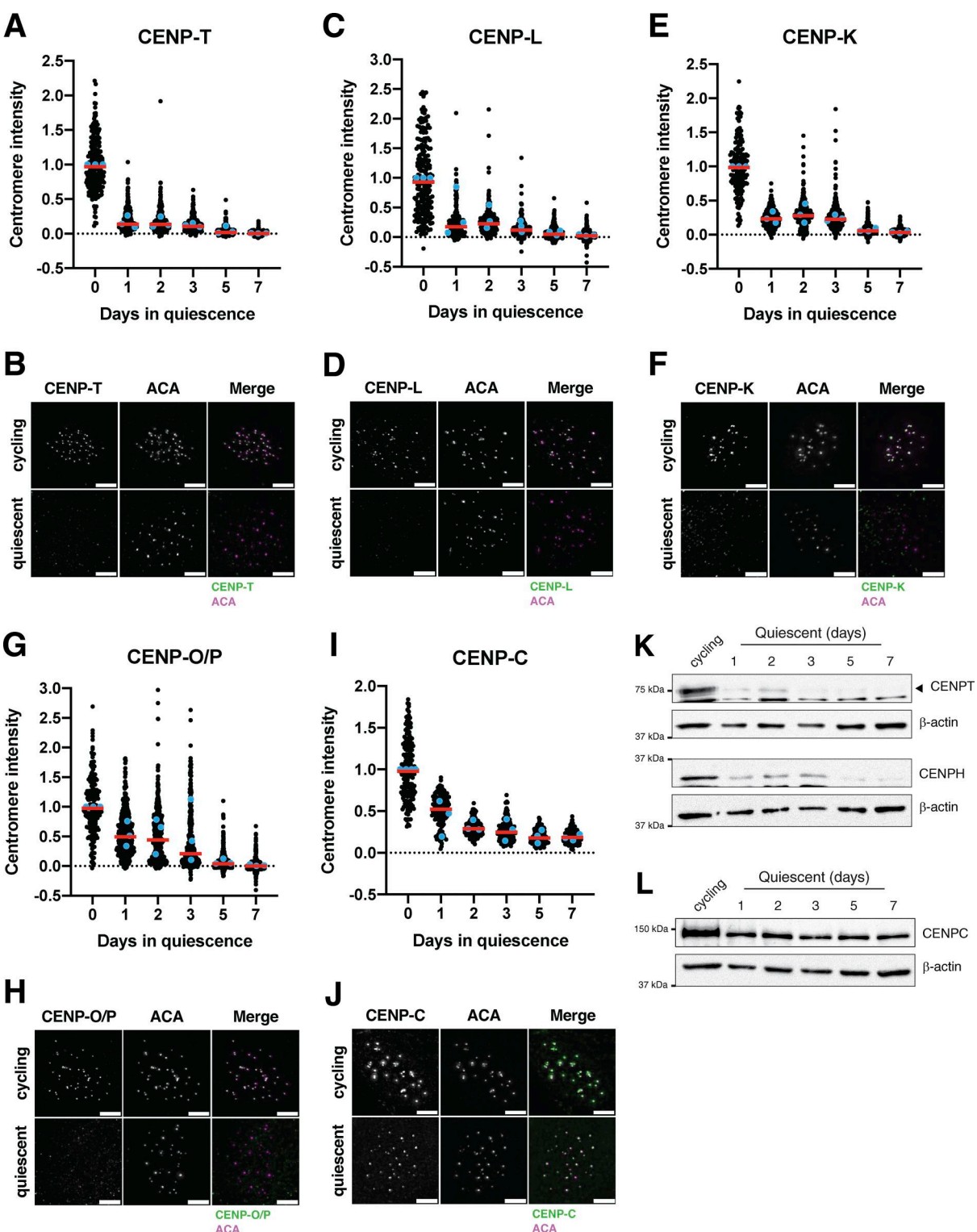

Figure 1.   **The centromere is rapidly disassembled upon quiescence entry. (A)** Graph showing CENP-T centromere intensity level over time of quiescence entry. Each point indicates the average centromere intensity level for all centromeres in a single cell, adjusted for background. Intensity values were normalized to day 0. Red line represents the median; blue points represent average of each replicate. Points were aggregated from three replicates. $n$ = 224, 392, 386, 419, 310, and 410 cells for 0, 1, 2, 3, 5, and 7-day time points, respectively. P values can be found in Table S2. **(B)** Representative immunofluorescence images of a cycling cell and cell in quiescence for 7 days. Cells were stained with CENP-T and anti-centromere (ACA) antibodies. Scale bar = 5 µm. **(C)** Graph showing CENP-L centromere intensity level over time of quiescence entry. Each point indicates the average centromere intensity level for all centromeres in a single cell, adjusted for background. Intensity values were normalized to day 0. Red line represents the median, and blue points represent average of each replicate. Points were aggregated from three replicates. $n$ = 224, 300, 314, 288, 281, and 290 cells for 0, 1, 2, 3, 5, and 7-day time points, respectively. P values can be found in Table S2. **(D)** Representative immunofluorescence images of a cycling cell and cell in quiescence for 7 days. Cells were stained with CENP-L and anti-centromere

(ACA) antibodies. Scale bar = 5 µm. **(E)** Graph showing CENP-K centromere intensity level over time of quiescence entry. Each point indicates the average centromere intensity level for all centromeres in a single cell, adjusted for background. Intensity values were normalized to day 0. Red line represents the median, and blue points represent average of each replicate. Points were aggregated from three replicates. n = 162, 442, 482, 525, 470, and 355 cells for 0, 1, 2, 3, 5, and 7-day time points, respectively. P values can be found in Table S2. **(F)** Representative immunofluorescence images of a cycling cell and cell in quiescence for 7 days. Cells were stained with CENP-K and anti-centromere (ACA) antibodies. Scale bar = 5 µm. **(G)** Graph showing CENP-O/P centromere intensity level over time of quiescence entry. Each point indicates the average centromere intensity level for all centromeres in a single cell, adjusted for background. Intensity values were normalized to day 0. Red line represents the median, and blue points represent average of each replicate. Points were aggregated from three replicates. n = 181, 522, 514, 465, 467, and 454 cells for 0, 1, 2, 3, 5, and 7-day time points respectively. P values can be found in Table S2. **(H)** Representative immunofluorescence images of a cycling cell and cell in quiescence for 7 days. Cells were stained with CENP-O/P and anti-centromere (ACA) antibodies. Scale bar = 5 µm. **(I)** Graph showing CENP-C centromere intensity level over time of quiescence entry. Each point indicates the average centromere intensity level for all centromeres in a single cell, adjusted for background. Intensity values were normalized to day 0. Red line represents the median, and blue points represent average of each replicate. Points were aggregated from three or more replicates. n = 201, 174, 169, 166, 196, and 194 cells for 0, 1, 2, 3, 5, and 7-day time points, respectively. P values can be found in Table S2. **(J)** Representative immunofluorescence images of a cycling cell and cell in quiescence for 7 days. Cells were stained with CENP-C and anti-centromere (ACA) antibodies. Scale bar = 5 µm. **(K)** Western blots of cells in quiescence for the indicated amount of days. Blots were incubated with CENP-T and CENP-H antibodies. β-Actin is used as a loading control. **(L)** Western blot of cells in quiescence for the indicated amount of days. Blot was incubated with CENP-C antibody. β-Actin is used as a loading control. Source data are available for this figure: SourceData F1.

transcription factors could control centromere protein gene expression during quiescence. The transcription factor FOXM1 plays a critical role in the regulation of cell cycle–related genes (Grant et al., 2013; Laoukili et al., 2005; Thiru et al., 2014). To determine whether FOXM1 could regulate centromere protein gene expression during quiescence, we analyzed available expression and transcription factor–binding data (Chen et al., 2013; ENCODE Project Consortium, 2012; Grant et al., 2013; Sanders et al., 2013; Thiru et al., 2014). We found that FOXM1 expression was strongly correlated with the expression of most centromere genes (Fig. 2 K and Fig. S2 D), with a clear enrichment of FOXM1 at centromere promoters based on ChIP-seq data (Chen et al., 2013; ENCODE Project Consortium, 2012; Grant et al., 2013; Sanders et al., 2013; Thiru et al., 2014) (Fig. 2 L). However, FOXM1 expression was not correlated with CENP-C, with the absence of clear promoter enrichment (Chen et al., 2013; ENCODE Project Consortium, 2012; Grant et al., 2013; Sanders et al., 2013; Thiru et al., 2014) (Fig. 2, K and L; and Fig. S2 D), and CENP-C was previously found to escape FOXM1 regulation (Khurana et al., 2024). In contrast to other centromere proteins, which are downregulated in quiescent cells, CENP-C shows a unique behavior and is upregulated during quiescence (Fig. 2, A and B). Importantly, we found that FOXM1 is highly downregulated in quiescent cells in our RNA-seq data (~30-fold) (Fig. S2 A). Thus, the downregulation of the FOXM1 transcription factor in quiescent cells could regulate the coordinated decrease in centromere protein expression while not affecting CENP-C mRNA levels.

## CENP-C retention supports centromere identity in quiescent cells

Because CENP-C was the only CCAN component to show continued transcription, protein abundance, and centromere localization in quiescent cells (Fig. 1, I, J, and L; and Fig. 2, A and B), we next considered whether its retention could play a role in maintaining centromere identity during quiescent arrest. We previously found that the continuous replenishment of CENP-A during quiescence was required for reentry into the cell cycle and that this deposition was dependent on HJURP and the Mis18 complex (Swartz et al., 2019). As CENP-C is necessary for CENP-A deposition in cycling cells and is important for HJURP and

MIS18 targeting (Falk et al., 2015; Moree et al., 2011; Stellfox et al., 2013; Westhorpe et al., 2015), we tested whether depleting CENP-C would compromise CENP-A incorporation in quiescent cells. To do so, we depleted CENP-C using RNAi and monitored CENP-A levels by immunofluorescence using a Halo-Tag pulse-chase system (Los et al., 2008) (Fig. S2, E and F; see Materials and methods). Although total centromeric CENP-A levels were unchanged (4% reduction; Fig. S2 G), the incorporation of new CENP-A was reduced upon loss of CENP-C (Fig. S2, H and I). CENP-C depletion led to a 15% reduction in CENP-A deposition over the course of 6 days (Fig. S2, H and I). As cells can remain quiescent for years, even this smaller reduction in CENP-A deposition in the absence of CENP-C could lead to the loss of CENP-A over longer periods of quiescent arrest. Previous work has also suggested the presence of CENP-C–independent CENP-A deposition pathways that could function in parallel (French et al., 2017; Hori et al., 2017). The lack of a complete abrogation of CENP-A deposition following CENP-C loss could reflect the use of such alternative pathways during quiescence or could also result from incomplete depletion of CENP-C in the first few days of siRNA treatment.

## *De novo* CENP-A deposition during quiescence exit occurs following the first mitosis

We next sought to analyze centromere dynamics during cell cycle reentry following quiescence exit. In particular, we considered when new CENP-A deposition would occur. In cycling cells, new CENP-A deposition at centromeres is regulated temporally, occurring once per cell cycle in early G1 (Bodor et al., 2013; Jansen et al., 2007; Stellfox et al., 2013). As quiescent cells enter G1 upon the return to proliferation, we hypothesized that new CENP-A deposition could occur either immediately upon release at the first G1, noncanonically at a later stage in the cell cycle, or during the subsequent G1 following the first mitotic division. In quiescent cells, CENP-A is retained at the centromeres but with only slightly reduced levels (~15% reduction) (Fig. S1 D). To visualize CENP-A deposition, we monitored a GFP-tagged CENP-A cell line (Su et al., 2016) by live-cell imaging after exit from quiescence (Fig. 3 A). Cells released from quiescence for 12 h did not have significantly different levels of GFP-CENP-A at the centromeres relative to quiescent cells (~60% reduction in

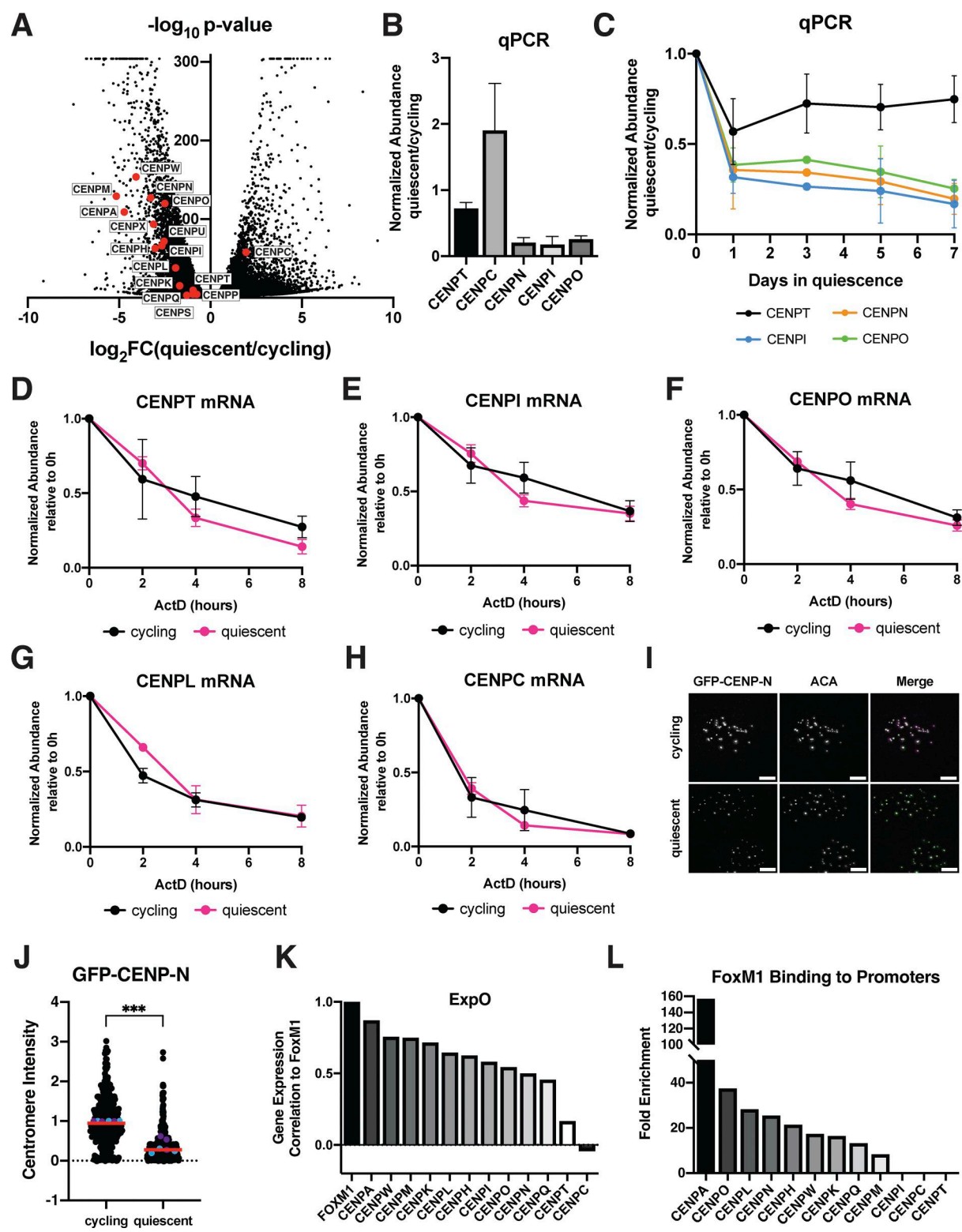

Figure 2. **Quiescent cells regulate centromere behavior through a transcriptional program. (A)** Volcano plot comparing mRNA abundances in quiescent and cycling RPE1 cells as measured by RNA sequencing. Centromere components are highlighted in red. A P value cutoff was imposed at P = 6.84E–305 for genes with P values of 0. Genes with low read counts (total counts <50) were excluded. **(B)** Graph showing the fold change in mRNA abundance between cycling cells and cells in quiescence for 7 days for the indicated centromere component as quantified by qPCR. CT values were normalized to those of GAPDH before comparing quiescent and cycling values. Graph shows at least three biological replicates, with three technical replicates each for each centromere mRNA. Bars represent mean ± SD. **(C)** Graph showing mRNA abundance for centromere components over time as cells enter quiescence. Fold change is calculated for each quiescent time point by dividing by cycling value. CT values were normalized to those of GAPDH before comparing quiescent and cycling values. Graph shows at least three biological replicates, with three technical replicates each for each centromere mRNA. Bars represent mean ± SD. **(D)** Graph showing CENPT mRNA

abundance over time after addition of 5 µg/ml actinomycin D in cycling and quiescent cells. Fold change is calculated for each ActD time point by dividing by the untreated (0 h) value for each respective condition (quiescent or cycling). CT values were normalized to those of GAPDH before comparing treated and untreated values. Graph shows three biological replicates, with three technical replicates each. Bars represent mean ± SD. **(E)** Graph showing CENPI mRNA abundance over time after addition of 5 µg/ml actinomycin D in cycling and quiescent cells. Fold change is calculated for each ActD time point by dividing by the untreated (0 h) value for each respective condition (quiescent or cycling). CT values were normalized to those of GAPDH before comparing treated and untreated values. Graph shows three biological replicates, with three technical replicates each. Bars represent mean ± SD. **(F)** Graph showing CENPO mRNA abundance over time after addition of 5 µg/ml actinomycin D in cycling and quiescent cells. Fold change is calculated for each ActD time point by dividing by the untreated (0 h) value for each respective condition (quiescent or cycling). CT values were normalized to those of GAPDH before comparing treated and untreated values. Graph shows three biological replicates, with three technical replicates each. Bars represent mean ± SD. **(G)** Graph showing CENPL mRNA abundance over time after addition of 5 µg/ml actinomycin D in cycling and quiescent cells. Fold change is calculated for each ActD time point by dividing by the untreated (0 h) value for each respective condition (quiescent or cycling). CT values were normalized to those of GAPDH before comparing treated and untreated values. Graph shows three biological replicates, with three technical replicates each. Bars represent mean ± SD. **(H)** Graph showing CENPC mRNA abundance over time after addition of 5 µg/ml actinomycin D in cycling and quiescent cells. Fold change is calculated for each ActD time point by dividing by the untreated (0 h) value for each respective condition (quiescent or cycling). CT values were normalized to those of GAPDH before comparing treated and untreated values. Graph shows three biological replicates, with three technical replicates each. Bars represent mean ± SD. **(I)** Representative immunofluorescence images of cells ectopically expressing GFP-CENP-N in cycling conditions or quiescence. Cells were stained for GFP (GFP-CENP-N) and anti-centromere antibody (ACA). Scale bar = 5 µm. **(J)** Graph showing GFP-CENP-N centromere intensity level in cycling and quiescent cells. Results were aggregated for two GFP-CENP-N–overexpressing cell line clones. Each point indicates the average centromere intensity level for all centromeres in a single cell, adjusted for background. Intensity values were normalized to cycling. Red line represents the median, and blue and purple points represent average of each replicate for each clone. Points were aggregated from three replicates for each. *n* = 398 and 438 cells for cycling and quiescent, respectively. P = 0.0003. *** represents P < 0.001, using unpaired *t* test with Welch's correction. **(K)** Graph showing the correlations in the gene expression profiles of various centromere protein genes and FoxM1 across samples from the Expression Project for Oncology (expO) data set (GEO accession number GSE2109). Correlations are ordered from largest to smallest and obtained from Thiru et al. (2014). **(L)** Graph showing FoxM1 binding to centromere protein gene promoters. ChIP-seq signal (ENCODE Project Consortium, 2012) was analyzed for FoxM1 compared with a control IP for regions encompassing 1 kb upstream and 100 base pairs downstream of the transcriptional start site for kinetochore genes. Fold enrichment values were obtained from Thiru et al. (2014).

centromeric GFP localization) (Fig. 3, B and C). These levels persisted until cells underwent their first mitosis (Fig. 3, D–F). Following completion of the first mitosis and entry into the subsequent G1, centromeric GFP-CENP-A levels increased dramatically (Fig. 3 D). Thus, cells released from quiescence undergo their first CENP-A deposition event during the G1 following the first mitotic division, but not during the G1 immediately following quiescence exit.

We next wanted to understand what differences between a normal G1 and a G1 immediately following the release from quiescence could underlie the timing of CENP-A deposition. In cycling G1 cells, CDK1 inactivation upon mitotic exit allows CENP-A deposition to occur (Silva et al., 2012), and the inhibition of CDK activity can induce CENP-A deposition even outside of its normal deposition window (Silva et al., 2012; Stankovic et al., 2017). However, we found that cells exiting quiescence into G1 do not have detectable levels of cyclin B1, indicative of lack of CDK1/cyclin B1 activity (Fig. 3 G). Despite the absence of CDK activity, these cells did not deposit CENP-A (Fig. 3, B and C). A second key regulatory mechanism that initiates CENP-A deposition during a normal G1 is PLK1 activity (McKinley and Cheeseman, 2014). Using immunofluorescence, we observed strong PLK1 localization at centromeres in cycling cells in G1. However, quiescence-released cells in G1 did not have PLK1 localization (Fig. 3, H and I). As PLK1 is required for CENP-A deposition under normal conditions (McKinley and Cheeseman, 2014), this would prevent CENP-A deposition. Finally, we used qPCR to monitor the levels of CENP-A mRNA over time of quiescence exit (Fig. S3 G). We found that cells released from quiescence do not fully restore CENP-A mRNA levels until 48 h after reentering the cell cycle (Fig. S3 G). Thus, although low CDK activity creates a potentially permissive environment for CENP-A deposition during the G1 following quiescence exit, the absence of PLK1 and lower levels of

CENP-A expression restrict the timing of CENP-A deposition to the subsequent G1 following the first mitotic division.

## Centromere reassembly during quiescence exit occurs in S phase but is independent of DNA replication

Unlike CENP-A, most other CCAN components are completely lost during a quiescent arrest, resulting in centromere disassembly (Fig. 1, A–H). As these CCAN proteins are required for chromosome segregation, they must be reassembled during the return to proliferation. In cells released from contact inhibition into full-serum media, CCAN subunit localization and protein levels were rapidly regained starting within 24 h of quiescence release (Fig. S3, A–F and H). qPCR of cells exiting quiescence showed a concurrent rapid increase in centromere protein mRNA levels within 24 h of quiescence release (Fig. S3 G). This centromere protein expression at early time points following quiescence exit reached levels above that of cycling cells but subsequently stabilized following an extended return to growth (Fig. S3 G).

To test at which stage of the cell cycle CCAN proteins are redeposited at centromeres, we released cells from quiescence for 24 h in the presence of EdU, a nucleotide analog that can be used to monitor progression through S phase via its incorporation during DNA replication. Cells that have entered S phase based on EdU staining regained centromeric levels of CENP-T and CENP-O/P, whereas EdU-negative cells did not (Fig. 4, A–F). In addition, the amount of EdU incorporation was closely correlated with CENP-T and CENP-O/P centromere intensity (Fig. 4, C and F). By contrast, CENP-C intensity was not correlated to EdU intensity and CENP-C levels increased in both EdU-positive and EdU-negative cells, although CENP-C showed a slightly greater relocalization in EdU-positive cells (Fig. S3, I and J).

Given the correlation between centromere reassembly and EdU incorporation, we next evaluated whether CCAN deposition

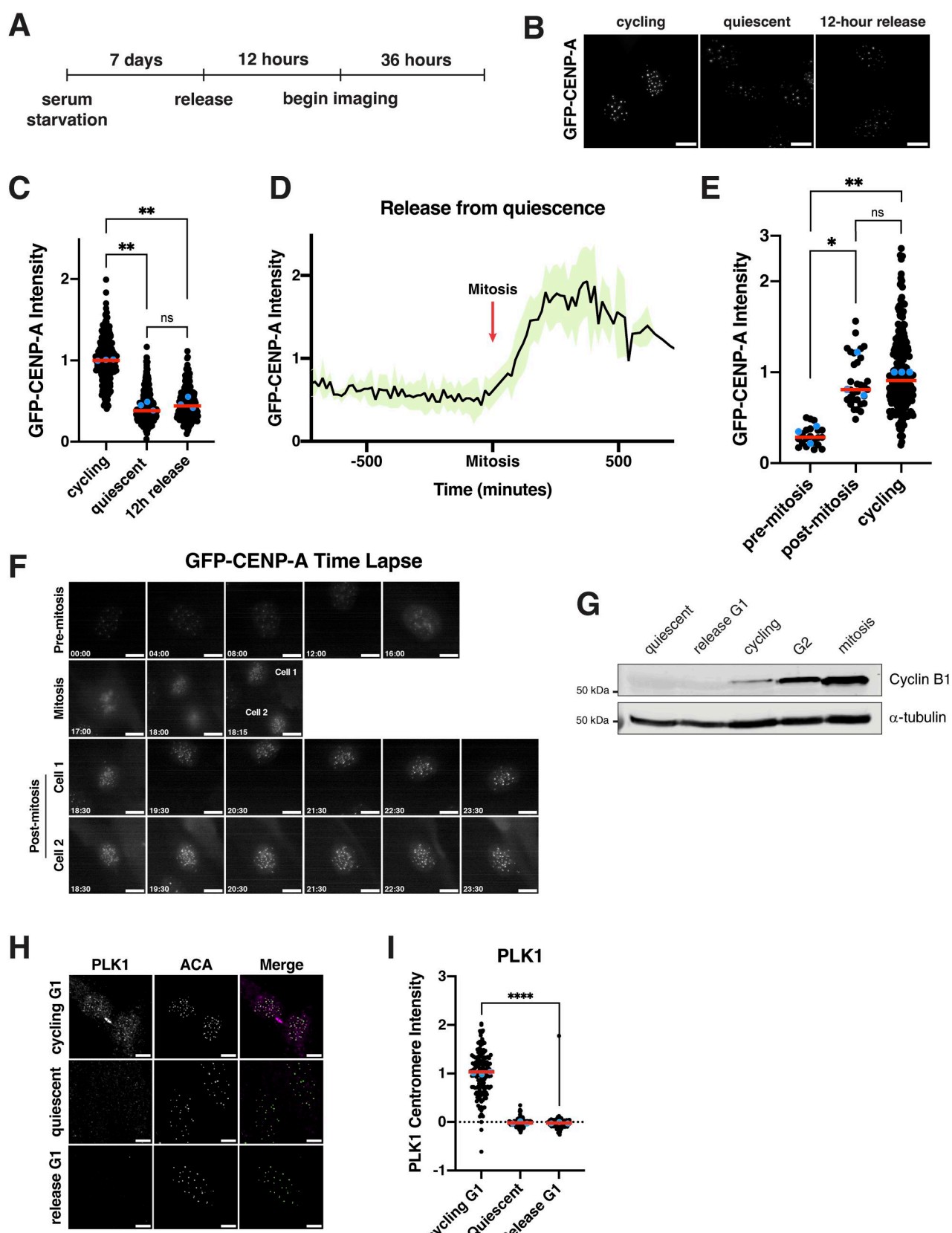

Figure 3.   ***De novo* CENP-A deposition during quiescence exit occurs following the first mitosis. (A)** Diagram showing experimental conditions for GFP-CENP-A quiescence exit live imaging experiments from Fig. 3, D–F. **(B)** Representative images of live GFP-CENP-A–expressing cells imaged for GFP from three different conditions: asynchronous cycling, 7 days of quiescence, or 12 h after release from quiescence. Scale bar = 10 μm **(C)** Graph quantifying cells from experiment in Fig. 3 B. GFP centromere intensity was measured in GFP-CENP-A–expressing cells from different conditions: asynchronous cycling, 7 days of

quiescence, or 12 h after release from quiescence. Cells were imaged live for GFP. Each point indicates the average centromere intensity level for 10 centromeres in a single cell, adjusted for background. Red line represents the median, and blue points represent averages for each replicate. Three replicates were conducted. Intensity values are normalized to cycling. P = 0.0054 between cycling and quiescent, P = 0.0026 between cycling and 12 h release, and P = 0.5430 between quiescent and 12 h release. ** represents P < 0.01; ns is not significant. n = 232, 256, 237 cells for cycling, quiescent, and 12 h release, respectively. P values calculated using unpaired t test with Welch's correction. **(D)** Graph showing the normalized level of GFP-CENP-A intensity at the centromeres as cells exit quiescence. Red arrow indicates time of mitosis. X-axis spans from 12 h before and 12 h after mitosis. Black line shows the means, and green shows mean ± SD. Data are aggregated from 33 cells quantified over 15-min or 20-min timeframes from 3 separate biological replicates. **(E)** Graph showing GFP-CENP-A intensity at the centromeres for cells released from quiescence before and after their first mitosis and for cycling cells (from Fig. 3 D, three replicates). Premitosis values are the average of all measurements from beginning of the live imaging or time of cell entry into imaging area to time of mitosis. Post-mitosis values are the average of all measurements from 1 h after the end of mitosis to the end of the time course or exit of the cell from the imaging area. Cycling values are from asynchronous control cells from the last imaging time frame. Each point indicates the average centromere intensity level for 10 centromeres or max number visible in a single cell, adjusted for background. Red line represents the median, and blue points are the average of each replicate. Intensity values are normalized to cycling. * represents P < 0.05, ** represents P < 0.01, and ns = not significant. P = 0.0427 between before mitosis and after mitosis, P = 0.0068 between before mitosis and cycling, and P = 0.6628 between after mitosis and cycling. P values were calculated using unpaired t test with Welch's correction. **(F)** Live imaging stills of a representative GFP-CENP-A cell exiting quiescence. Entry into mitosis is indicated. After mitosis, both cells are indicated and shown. Time units are hours:minutes. **(G)** Western blot of 7-day quiescent cells, cells released for 16 h from quiescence (G1), asynchronous cycling cells, cells arrested in G2 using RO-336 (Vassiliev, 2006; Vassilev et al., 2006), and cells arrested in mitosis using S-trityl-L-cysteine (STLC) (Skoufias et al., 2006). Blot was incubated with cyclin B1 antibody, and α-tubulin is used as a loading control. **(H)** Representative immunofluorescence images of cycling cells in G1 (identified by presence of a midbody), 7-day quiescent cells, and cells released from quiescence for 16 h stained for PLK1, and anti-centromere (ACA) antibodies. Scale bar = 5 µm. **(I)** Graph representing PLK1 intensity levels at the centromere for experiment from Fig. 3 H. Each point indicates the average centromere intensity level for all centromeres in a single cell, adjusted for background. Intensity values were normalized to cycling G1. Red line represents the median, and blue points represent average of each replicate. **** represents P < 0.0001, using unpaired t test with Welch's correction. Source data are available for this figure: SourceData F3.

upon quiescence exit requires entry into S phase. To test this, we released quiescent cells into the cell cycle for 48 h in the presence of palbociclib, a CDK4/6 inhibitor that prevents S phase entry and induces a G1 arrest. Cells treated with palbociclib did not enter S phase and no longer incorporated EdU (Fig. S3, K and L). In addition, palbociclib treatment prevented the redeposition of CENP-T and CENP-O/P, whereas CENP-C was still able to relocalize (Fig. 4, G–I; and Fig S3 M). Thus, preventing S phase entry upon quiescence exit prevents centromere reassembly.

To test whether centromere reassembly requires DNA replication, we next released cells from quiescence for 48 h in the presence or absence of thymidine, which prevents DNA replication and arrests cells in early S phase (Pedeux et al., 1998; Wang and Wang, 2022). Cells released from quiescence in the presence of thymidine were able to regain CENP-T, CENP-O/P, and CENP-C protein localization (Fig. 4, J–L; and Fig. S3 N). Thus, centromere deposition does not require DNA replication. In addition, this experiment confirms that CCAN deposition indeed occurs during S phase and does not require entry into a later cell cycle stage.

Finally, we tested whether lack of redeposition of certain CCAN components prior to S phase was a result of lack of expression of those proteins or whether those proteins were present but incapable of being recruited to centromeres. Whereas cells released from quiescence for 48 h in the presence of thymidine regained cycling protein levels of CENP-T and CENP-H, those released in the presence of palbociclib did not (Fig. S3 O). Conversely, CENP-C protein, which relocalizes earlier than CENP-T or CENP-O/P in the presence of palbociclib, is present in cells released from quiescence for 48 h in all conditions (Fig. S3 O). Thus, protein levels could help explain the relocalization behavior of CENP proteins in released cells.

Although untreated cells released from quiescence recovered normal cycling levels of CCAN at the centromeres, we observed even higher centromeric CCAN protein intensity in thymidine-treated cells relative to untreated cells (Fig. 4, K and L). This

suggests that cells can restore homeostatic levels of CENP-T and CENP-O/P upon the return to proliferation but that extending the time spent in S phase can lead to additional centromere protein deposition. Indeed, prolonging S phase even in cycling cells through 48 h thymidine incubation also resulted in increased centromere protein deposition (Fig. S3, P and Q). Thus, cells regulate the appropriate level of centromere protein deposition, but prolonging the stage at which deposition occurs and access to potential deposition factors can disturb this regulatory control. Together, these results show that centromere reassembly following exit from quiescence occurs in S phase and that this deposition is independent of DNA replication.

### Dynamics of centromere disassembly and re-assembly

Quiescence is a unique cellular state that requires a cell to enforce a prolonged arrest while maintaining its capacity to divide again in the future. Prior work on cell division components has focused primarily on cycling cells. However, understanding the logic of cell division in an intact organism requires a consideration of the diverse physiological circumstances in which cells exist, including nondividing states. Here, we show how cells modify a key cellular structure, the centromere, to meet the demands of the quiescent state. In proliferating cells, centromere components are constitutively localized to the centromere during all stages of the cell cycle. By contrast, we show that quiescence provides a unique context in which the centromere is largely disassembled through the complete loss of specific centromere components, such as CENP-T, CENP-K, CENP-L, and CENP-O/P. This loss is regulated in part by a rapidly enacted transcriptional program, which downregulates centromere component transcription within 24 h of serum starvation. However, quiescent cells must also preserve the centromere to be able to segregate their chromosomes upon return to the cell cycle. We show that selected centromere components required for maintaining centromere identity, such as CENP-A and CENP-C, are retained. Thus, the centromere is a prime example

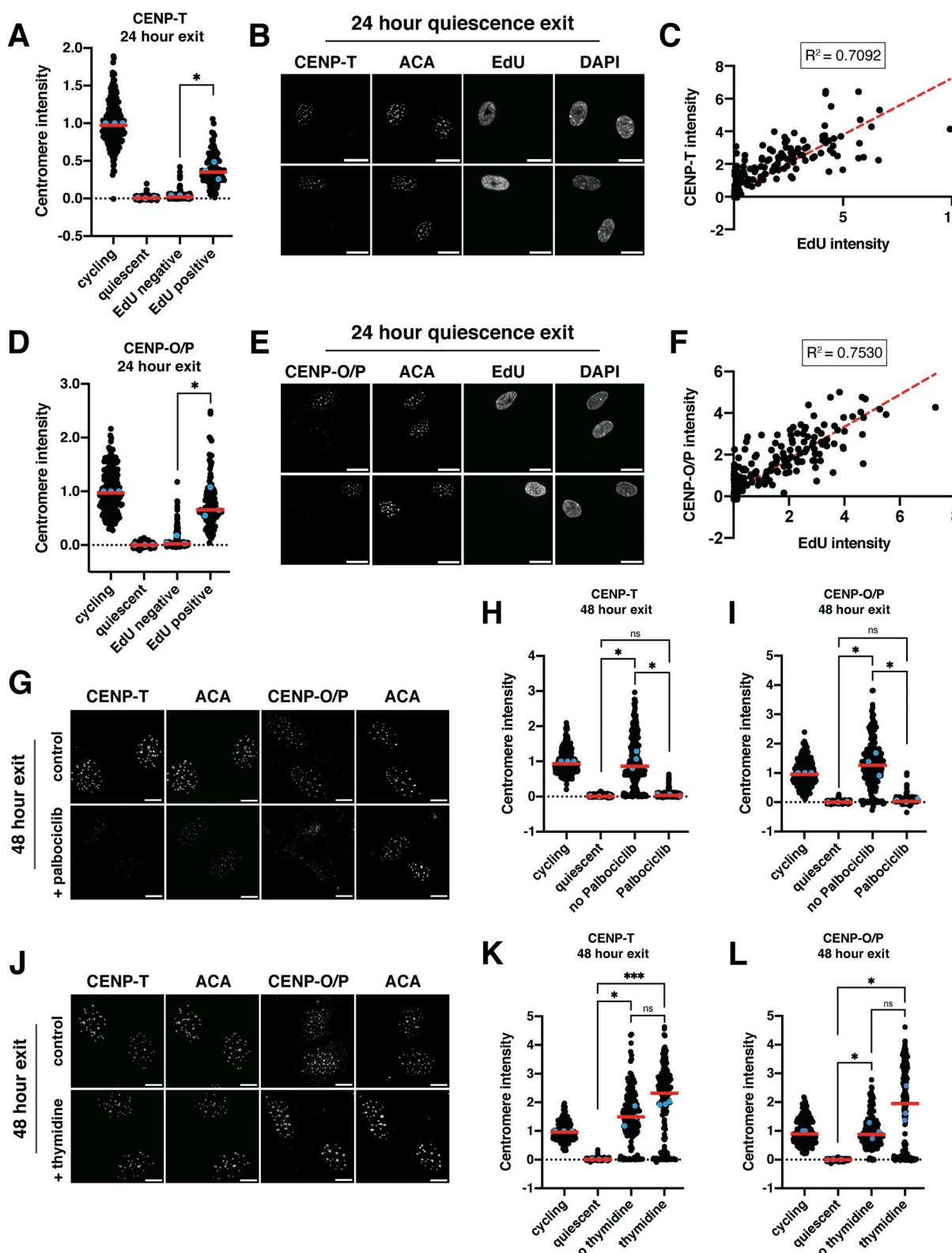

Figure 4. **Centromere reassembly during quiescence exit occurs in S phase but is independent of DNA replication. (A)** Graph showing CENP-T centromere intensity levels for different conditions: cycling, 7 days of quiescence, and either EdU-negative or EdU-positive cells from cells released 24 h from quiescence. Each point indicates the average centromere intensity level for all centromeres in a single cell, adjusted for background. Intensity values were normalized to cycling. Red line represents the median, and blue points represent average of each replicate. Points were aggregated from three replicates. $n$ = 202, 336, 202, and 107 cells for cycling, quiescent, 24 h release EdU negative, and 24 h release EdU positive, respectively. * represents $P < 0.05$; $P = 0.0348$ between EdU negative and EdU positive using unpaired $t$ test with Welch's correction. Other pairwise comparisons can be found in Table S2. **(B)** Representative immunofluorescence images of cells released 24 h from quiescence. EdU-positive and -negative cells are shown. Cells were stained with CENP-T and

anti-centromere (ACA) antibodies. Scale bar = 20 µm. **(C)** Graph showing CENP-T centromere intensity plotted against EdU intensity for each cell. CENP-T intensity is the average CENP-T centromere intensity level for all centromeres in a cell, adjusted for background. EdU intensity is the mean EdU intensity in the nucleus of the same cell. Points were aggregated from three replicates. Values were normalized within each replicate. $R^2$ = 0.7092. $n$ = 309 cells. **(D)** Graph showing CENP-O/P centromere intensity levels for different conditions: cycling, 7 days of quiescence, and either EdU-negative or EdU-positive cells from cells released 24 h from quiescence. Each point indicates the average centromere intensity level for all centromeres in a single cell, adjusted for background. Intensity values were normalized to cycling. Red line represents the median, and blue points represent average of each replicate. Points were aggregated from three replicates. $n$ = 214, 389, 185, and 114 cells for cycling, quiescent, 24 h release EdU negative, and 24 h release EdU positive, respectively. * represents $P < 0.05$; $P$ = 0.0424 between EdU negative and EdU positive using unpaired $t$ test with Welch's correction. Other pairwise comparisons can be found in Table S2. **(E)** Representative immunofluorescence images of cells released 24 h from quiescence. EdU-positive and -negative cells are shown. Cells were stained with CENP-O/P and anti-centromere (ACA) antibodies. Scale bar = 20 µm. **(F)** Graph showing CENP-O/P centromere intensity plotted against EdU intensity for each cell. CENP-O/P intensity is the average CENP-O/P centromere intensity level for all centromeres in a cell, adjusted for background. EdU intensity is the mean EdU intensity in the nucleus of the same cell. Points were aggregated from three replicates. Values were normalized within each replicate. $R^2$ = 0.753. $n$ = 298 cells. **(G)** Representative immunofluorescence images of cells released from quiescence for 48 h with or without addition of 1 µM palbociclib. Cells were stained for either CENP-T or CENP-O/P and anti-centromere (ACA) antibodies. Scale bar = 10 µm. **(H)** Graph showing CENP-T centromere intensity for different conditions: cycling, quiescent, 48 h release from quiescence without palbociclib, and 48 h release from quiescence with palbociclib. Each point indicates the average centromere intensity level for all centromeres in a single cell, adjusted for background. Intensity values were normalized to cycling. Red line represents the median, and blue points represent average of each replicate. Points were aggregated from three replicates. $n$ = 353, 429, 237, and 280 for cycling, quiescent, no palbociclib, and palbociclib, respectively. * represents $P < 0.05$; ns = not significant. $P$ = 0.0184 between palbociclib and no palbociclib, $P$ = 0.0177 between no palbociclib and quiescent, and $P$ = 0.0878 between palbociclib and quiescent, using unpaired $t$ test with Welch's correction. Other pairwise comparisons can be found in Table S2. **(I)** Graph showing CENP-O/P centromere intensity for different conditions: cycling, quiescent, 48 h release from quiescence without palbociclib, and 48 h release from quiescence with palbociclib. Each point indicates the average centromere intensity level for all centromeres in a single cell, adjusted for background. Intensity values were normalized to cycling. Red line represents the median, and blue points represent average of each replicate. Points were aggregated from three replicates. $n$ = 244, 355, 215, and 256 for cycling, quiescent, no palbociclib, and palbociclib, respectively. * represents $P < 0.05$, ns = not significant. $P$ = 0.0286 between palbociclib and no palbociclib; $P$ = 0.0278 between no palbociclib and quiescent, and $P$ = 0.1902 between palbociclib and quiescent, using unpaired t test with Welch's correction. Other pairwise comparisons can be found in Table S2. **(J)** Representative immunofluorescence images of cells released from quiescence for 48 h with or without addition of 2 mM thymidine. Cells were stained for either CENP-T or CENP-O/P and anti-centromere (ACA) antibodies. Scale bar = 10 µm. **(K)** Graph showing CENP-T centromere intensity for different conditions: cycling, quiescent, 48 h release from quiescence without thymidine, and 48 h release from quiescence with thymidine. Each point indicates the average centromere intensity level for all centromeres in a single cell, adjusted for background. Intensity values were normalized to cycling. Red line represents the median, and blue points represent average of each replicate. Points were aggregated from three replicates. $n$ = 247, 343, 112, and 166 for cycling, quiescent, no thymidine, and thymidine, respectively. * represents $P < 0.05$; *** represents $P < 0.001$; ns = not significant. $P$ = 0.1546 between thymidine and no thymidine, $P$ = 0.0176 between no thymidine and quiescent, and $P$ = 0.0002 between thymidine and quiescent, using unpaired $t$ test with Welch's correction. Other pairwise comparisons can be found in Table S2. **(L)** Graph showing CENP-O/P centromere intensity for different conditions: cycling, quiescent, 48 h release from quiescence without thymidine, and 48 h release from quiescence with thymidine. Each point indicates the average centromere intensity level for all centromeres in a single cell, adjusted for background. Intensity values were normalized to cycling. Red line represents the median, and blue points represent average of each replicate. Points were aggregated from three replicates. $n$ = 260, 328, 149, and 190 for cycling, quiescent, no thymidine, and thymidine, respectively. * represents $P < 0.05$; ns = not significant. $P$ = 0.1301 between thymidine and no thymidine, $P$ = 0.0250 between no thymidine and quiescent, and $P$ = 0.0368 between thymidine and quiescent, using unpaired $t$ test with Welch's correction. Other pairwise comparisons can be found in Table S2.

of how cells modify and regulate a cellular structure to enable the requirements of quiescence.

Release from quiescence and return to growth provides an opportunity to study centromere reassembly and centromere protein deposition. Whereas previous work studied centromere protein deposition in the context of cycling cells (Prendergast et al., 2016), where preexisting pools of protein can be difficult to distinguish from newly deposited protein, quiescence exit allows for the observation of completely *de novo* centromere protein deposition. We show here that CENP-A deposition in cells that exit quiescence occurs in the G1 following the first mitosis and not in the G1 immediately following quiescence release. Although both states lack CDK/cyclin B1 activity and would therefore be expected to be permissive for CENP-A deposition, we find that cells in the G1 following quiescence release additionally lack PLK1 localization to centromeres. This molecular difference between a normal cycling G1 and a quiescence release G1 could explain the differences in CENP-A deposition behavior between these two otherwise similar G1-like cell cycle phases. In addition, we find that new CENP-T and CENP-O/P are deposited *de novo* at S phase following quiescence release, independent of DNA replication, whereas increases in CENP-C levels can occur even earlier. This CENP-C assembly prior to S phase entry occurs

before full recovery of CENP-A levels, possibly because CENP-C recruitment can still occur in nonstoichiometric ratios to CENP-A due to its two distinct CENP-A–binding domains and its ability to self-dimerize (Hara et al., 2023; Kato et al., 2013). Early CENP-C recruitment is also made possible by the presence of CENP-C protein, whereas other CCAN proteins are not expressed until entry into S phase (Fig. S3 O).

Here, we find that centromere dynamics during quiescence entry and exit are partly regulated at the gene expression level. Within 24 h of quiescence entry, CCAN protein mRNAs decrease dramatically, coinciding with the loss of these proteins at the centromeres. This transcriptional downregulation of centromere protein levels is essential for centromere disassembly during quiescence, as the ectopic expression of certain CCAN proteins, such as CENP-N, led to the retention of CENP-N at the centromeres. Thus, loss of CENP-N centromere localization in quiescence relies on its transcriptional downregulation. However, the re-expression of a single centromere protein would, in most cases, not be enough to promote centromeric relocalization in quiescent cells. We chose to ectopically express GFP-CENP-N, as CENP-N is known to bind to CENP-A and CENP-C (Carroll et al., 2009; Carroll et al., 2010; McKinley et al., 2015), both of which remain present at the centromere during quiescence. For

other CCAN proteins, the interconnected nature of centromere protein localization would likely prevent the relocalization of an individual CCAN protein in quiescent cells even upon ectopic expression (McKinley et al., 2015). Similarly to quiescence entry, 24 h of quiescence exit leads to a large spike in centromere protein gene expression, which reaches levels above that of cycling cells. Although these mRNA levels eventually stabilize after extended quiescence release, the initial spike in CCAN gene expression could be beneficial for producing a large amount of protein available for centromere reassembly.

Despite strong regulation at the gene expression level, other mechanisms may also contribute to centromere dynamics during quiescence. For example, CENP-T mRNA levels decrease by only ~30% in quiescent human cells, and CENP-T expression increases in quiescent mouse cells (Fig. 2, B and C; and Fig. S2 B). However, quiescent human cells have no CENP-T protein present, and there is no CENP-T at centromeres in either organism. Thus, other mechanisms, such as protein degradation or other posttranslational mechanisms, could also regulate CENP-T protein levels.

In all, this work not only highlights important paradigms in the regulation of cellular structures during quiescence but also uncovers key aspects of centromere biology within a novel cellular context.

## Materials and methods

### Cell lines
hTERT RPE-1 and NIH-3T3 cell lines were cultured in DMEM supplemented with either 100 U/ml penicillin and streptomycin and 2 mM L-glutamine with (growth) or without (serum starvation) 10% FBS. Cells were grown at 37°C with 5% $CO_2$. hTERT RPE-1 cells are hTERT-immortalized retinal pigment epithelial cells of female origin. The NIH-3T3 cell line is derived from a male mouse embryo. GFP-CENP-A cell line was previously described (Su et al., 2016). GFP-CENP-N cell line was generated by lentiviral transduction. Lentivirus was made by transfection with X-tremegene-9 (Roche) of GFP-CENP-N lentiviral plasmid, VSV-G envelope plasmid, and psPAX2 (#12260; Addgene plasmid) packaging plasmid into HEK-293T cells. After transduction of RPE1 cells, GFP-positive cells were then sorted by fluorescence-activated cell sorting, and monoclonal cell lines were selected based on characteristic GFP localization by live imaging. Two separate clones are used in this study. Cells were tested regularly for mycoplasma contamination.

### Cell culture and reagents
For quiescence induction, cells were allowed to grow until reaching confluence. Cells were then left to contact inhibit for 1–2 days before serum starvation. For serum starvation, full serum media were removed from cells, which were then washed once with PBS before replacing with serum-free DMEM. Media were then changed every 24 h until cells were harvested. Quiescent cells were serum starved for 7 days unless otherwise noted. For quiescence release, quiescent cells were trypsinized or treated with PBS + 5 mM EDTA before replating at lower density in full serum (10% FBS) media for the indicated amounts of time.

Drugs used on cell lines were EdU (10 µM; Vector Laboratories), actinomycin D (5 µg/ml; Santa Cruz Biotechnology), thymidine (2 mM; Sigma-Aldrich), Palbociclib (1 µM; Selleck), Janelia Fluor HaloTag Ligand (125 nM or 30 nM; Promega), and RO-336 (6 µM; Millipore Sigma).

### Immunofluorescence
Cells were fixed with either 4% formaldehyde diluted in PBS + 0.5% Triton X-100 or in cold Methanol. After washing with PBS + 0.1% Triton X-100 and incubation in blocking buffer (20 mM Tris-HCl, 150 mM NaCl, 0.1% Triton X-100, 3% bovine serum albumin, and 0.1% NaN₃, pH 7.5) for 30 min or overnight, primary antibody was added for 1 h or overnight. Cells were then washed before addition of Cy2-, Cy3-, or Cy5-conjugated secondary antibodies (Jackson ImmunoResearch Laboratories) for 1 h, followed by 10 min of 1 µg/ml Hoechst-33342 (Sigma-Aldrich) diluted in 0.1% PBS-Tx to stain DNA. For cells with EdU staining, after incubation with secondary antibody, Click-iT buffer (100 mM Tris, pH 8.0, 1 mM $CuSO_4$, 5 µM Alexa Fluor azide (Life Technologies), and 100 mM ascorbic acid) was added for 30 min before washing and Hoechst staining. Coverslips were mounted onto slides with ProLong Gold antifade reagent (Invitrogen).

### Microscope image acquisition
Microscopy was conducted using DeltaVision Ultra High-Resolution microscope with a pco.edge cooled camera. All immunofluorescence images were acquired at room temperature. All immunofluorescence images shown in Fig. 1, Fig. 2, Fig. 4, Fig. S1, F and H, and Fig. S2 were taken with Olympus 60×/1.42, UPLXAP0 Objective, with oil, with ~30–40 Z-sections at 0.2-µm steps. Images in Fig. S1 A were taken on Olympus U-Plan Fluorite PH1 20×/0.45 w/Corr. Collar Objective. Acquisition software was AcquireUltra. Images were analyzed with Fiji (ImageJ, NIH) and CellProfiler (Carpenter et al., 2006). Images were deconvolved on the DeltaVision. A list of antibodies and their dilutions can be found in Table S1 (Dudka et al., 2025; Gascoigne et al., 2011; McKinley et al., 2015). Quantification of fluorescence intensity for immunofluorescence experiments was conducted on unprocessed, maximally projected images using FIJI/Image J or using a custom CellProfiler and Python pipeline (available on GitHub: https://github.com/oceanema/The-dynamics-of-centromere-assembly-and-disassembly-during-quiescence.git). These images were acquired using the same microscope and acquisition settings on the same day, unless otherwise noted in the figure legends.

### Live-cell imaging
For live-cell imaging, GFP-CENP-A–tagged cells were induced to enter quiescence by contact inhibition and 7 days of serum starvation. Quiescent cells were then released from quiescence and replated onto 12-well glass-bottom plates (Cellvis, P12-1.5P) and centrifuged to promote adherence. 1 h prior to imaging, the media were changed to $CO_2$-independent media (Gibco) supplemented with 10% FBS, 100 U/ml penicillin and streptomycin, and 2 mM L-glutamine. Cells were imaged 12 h after replating for 36 or more hours with 15-min time points. For the last replicate,

images were taken at 20-min time points, these values were merged to the nearest 5-min mark in Fig. 3 D. Imaging for GFP was conducted at 37°C on a Nikon eclipse microscope (40×). Images were analyzed with Fiji (ImageJ, NIH).

### Determining EdU-positive and -negative cells

Maximally projected images were processed by CellProfiler, where the EdU channel was measured within nuclear boundaries for each cell. To find the threshold EdU value to determine positive and negative cells, mean nuclear EdU intensities for all cells were used to fit a Gaussian Mixture Model (GMM) with two components. GMM was then used to determine threshold based on predictions from a range of values. Code was developed using help from ChatGPT and available on GitHub: (https://github.com/oceanema/The-dynamics-of-centromere-assembly-and-disassembly-during-quiescence.git).

### Halo-Tag pulse-chase

HaloTag-CENP-A cells were induced to enter quiescence by contact inhibition and serum starvation. 4 days after the start of serum starvation, cells were treated with control or CENP-C siRNAs. 24 h later, preexisting CENP-A was blocked by incubation with 125 nM of Halo JF549. After 2 days, siRNAs were reapplied. After 4 more days of quiescence (11 total days of quiescence), cells were fixed. During this time after blocking of preexisting CENP-A, newly synthesized, unblocked HaloTag-CENP-A is deposited at centromeres. Immunofluorescence was conducted as above; however, during incubation with the secondary, cells were also incubated with 30 nM Halo JF646 for 1 h before washing and Hoechst staining. We then compared total or newly deposited CENP-A between quiescent cells treated with CENP-C or control siRNAs by immunofluorescence.

### Western blot

Cells were lysed with urea lysis buffer (50 mM Tris, pH 7.5, 150 mM NaCl, 0.5% NP-40, 0.1% SDS, 6.5 M Urea, 1× Complete EDTA-free protease inhibitor cocktail [Roche], and 1 mM phenylmethylsulfonyl fluoride [PMSF]) on ice for 25 min, followed by centrifugation to remove cellular debris. Then, Laemmli sample buffer was added with 2-mercaptoethanol, and samples were heated at 95°C for 5 min. Samples were then loaded onto acrylamide gels for SDS-PAGE, followed by 1 h or overnight transfer onto nitrocellulose membranes. Membranes were incubated for 1 h in blocking buffer (2.5% milk in Tris-buffered saline + 0.1% Tween-20), then 1 h or overnight in primary antibodies, then 1 h in either HRP-conjugated secondary antibodies (GE Healthcare; Digital) or in IRDye secondary antibodies (LI-COR). For imaging, blots were either incubated in clarity-enhanced chemiluminescence substrate (Bio-Rad) and imaged using the KwikQuant Imager (Kindle Biosciences) or directly imaged using the Odyssey CLx Imager (LI-COR). β-Actin is used as a loading control. Antibodies are listed in Table S1.

### Flow cytometry analyses

For DNA content analysis, cells were harvested, resuspended in 1 ml of PBS, and fixed with addition of 9 ml of ice-cold ethanol. Cells were then resuspended in PBS to rehydrate, followed by incubation in blocking buffer (20 mM Tris-HCl, 150 mM NaCl, 0.1% Triton X-100, 3% bovine serum albumin, and 0.1% NaN₃, pH 7.5) for 30 min. After blocking, cells were resuspended in PBS containing 10 μg/ml RNase A and 20 μg/ml propidium iodide (PI, Invitrogen) and incubated for 30 min before analysis by flow cytometry. PI signal was measured on an LSRFortessa (BD Biosciences) flow cytometer. Results were analyzed with FlowJo software. Data were collected on at least 10,000 cells for each condition per experiment.

### siRNA treatment

Custom siRNAs against CENP-C (5′-GAACAGAAUCCAUCACAA AUU-3′) and a nontargeting control pool (D-001206-13) were obtained from Dharmacon. siRNAs were used at a final concentration of 50 nM. siRNAs and Lipofectamine RNAiMAX (Invitrogen) were mixed at equal volume in Opti-MEM Reduced Serum Medium (Thermo Fisher Scientific), vortexed, and allowed to incubate for 20 min before adding to cells. Media were changed 24 h later back to no serum media for quiescence.

### RNA isolation, reverse transcription, qPCR, and RNA sequencing

Cells were lysed in 400 μl of TRIzol RNA isolation reagent (Thermo Fisher Scientific) and frozen at –80°C or used directly for the next step. 120 μl of chloroform was added to samples, and tubes were vortexed vigorously for at least 30 s. After centrifuging at 4°C, the aqueous phase was mixed with an equal volume of chloroform. Samples were vortexed vigorously, centrifuged at 4°C, and the aqueous phase was again transferred to a new tube. GlycoBlue Coprecipitant (Invitrogen), 5 M NaCl, and an equal volume of isopropanol were added, and samples were incubated on dry ice for 30 min. After centrifugation, pellets were isolated, washed with 75% ethanol, and resuspended in water. All reagents were RNase free. Maxima First Strand cDNA Synthesis Kit for RT-qPCR (Thermo Fisher Scientific) was used for reverse transcription according to the manufacturer's instructions. For qPCR, cDNAs were diluted and mixed with 1 μM primers and 2× SYBR Green PCR Master Mix (Thermo Fischer Scientific) in 384-well plates. Three technical replicates per cDNA sample and primer pair were used. Primer sequences can be found in Table S1.

For RNA sequencing: rRNA depletion and library preparation were done with the KAPA RNA HyperPrep Kit with RiboErase (HMR) (KK8560; Roche) according to the manufacturer's instructions. Sequencing was performed on an Illumina NovaSeq SP. For analysis, reads were aligned to the human (GRCh38; gencode v25) genome using hisat2. Expression matrix was generated using featureCounts. Genes with low read count (<50) were filtered out. Differential expression analysis was conducted with DESeq2.

### Analysis of centromere protein gene expression and FoxM1

We used data previously analyzed by our lab (Thiru et al., 2014) to generate the graphs in Fig. 2, K and L; and Fig. S2 D. For gene expression correlation, the expression of centromere genes and FoxM1 were analyzed from two data sets: the Broad Cancer Cell Line Encyclopedia (Barretina et al., 2012) (CCLE), which

contains expression data for 991 cancer cell lines, and the Expression Project for Oncology data set (GEO accession number GSE2109), which contains data for 2,158 tumor samples. Correlations in the gene expression profiles of the kinetochore genes with FoxM1 were assessed through pairwise comparisons between the expression profiles for each pair of genes for both data sets. For FoxM1 binding to promoters, ChIP-seq signal (ENCODE Project Consortium, 2012) was analyzed for FoxM1 compared with a control IP for regions encompassing 1 kb upstream and 100 base pairs downstream of the transcriptional start site for kinetochore genes. For more details of the methods of computation analyses, see Thiru et al. (2014).

### Statistical analysis

Statistical analyses were performed using Prism (GraphPad Software). All statistics for pairwise comparison were conducted using unpaired, parametric, two-tailed, *t* test with Welch's correction. Data distribution was assumed to be normal (Gaussian), but this was not formally tested. For some graphs, we have chosen to keep the underlying raw data points while also adding colored points representing the average of each replicate (this is indicated in the figure legends). All statistics were calculated from these replicate averages. For other experiments, we show the experimental replicate averages, with the correspondingly calculated statistics. For graphs with multiple points of comparison, we display the statistics only for crucial comparisons for clarity. However, we calculated statistics for all of the pairwise comparisons, which are found in Table S2. Further details of statistical tests and sample sizes are provided in the figure legends.

### Online supplemental material

Fig. S1 related to Fig. 1 shows experiments validating quiescence entry, quantification of CENP-A centromere intensity during quiescence entry, mouse 3T3 experiments, and western blots controlling for antibody-banding patterns. Fig. S2 related to Fig. 2 shows controls for RNA sequencing. qPCR of 3T3 cells, control for ActD time course, analysis of CCLE data for FOXM1, and data for CENP-C siRNA experiments. Fig. S3 related to Fig. 4 shows quantification of centromere intensity immunofluorescence, qPCR, and western blots for return to growth, CENP-C data for EdU, palbociclib, and thymidine experiments, and controls for palbociclib and thymidine experiments. Table S1 shows antibody information and primer sequences used for qPCR. Table S2 shows statistics and P values for all pairwise comparisons between conditions for all figures. Statistics were calculated as described in the Materials and methods section.

## Data availability

Data are available in the article itself and its supplementary material or upon request.

## Acknowledgments

We thank members of the Cheeseman lab for feedback throughout the process. We would also like to thank members of the Whitehead Genome Technology Core.

This work was supported by a grant to I.M. Cheeseman from the National Institutes of Health/National Institute of General Medical Sciences (R35GM126930).

Author contributions: Océane Marescal: conceptualization, formal analysis, investigation, methodology, project administration, software, supervision, validation, visualization, and writing—original draft, review, and editing. Kuan-Chung Su: data curation. Brittania Moodie: investigation, methodology, and validation. Noah J.L. Taylor: investigation and validation. Iain M. Cheeseman: conceptualization, funding acquisition, supervision, and writing—review and editing.

Disclosures: The authors declare no competing interests exist.

Submitted: 8 September 2025

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

# Supplemental material

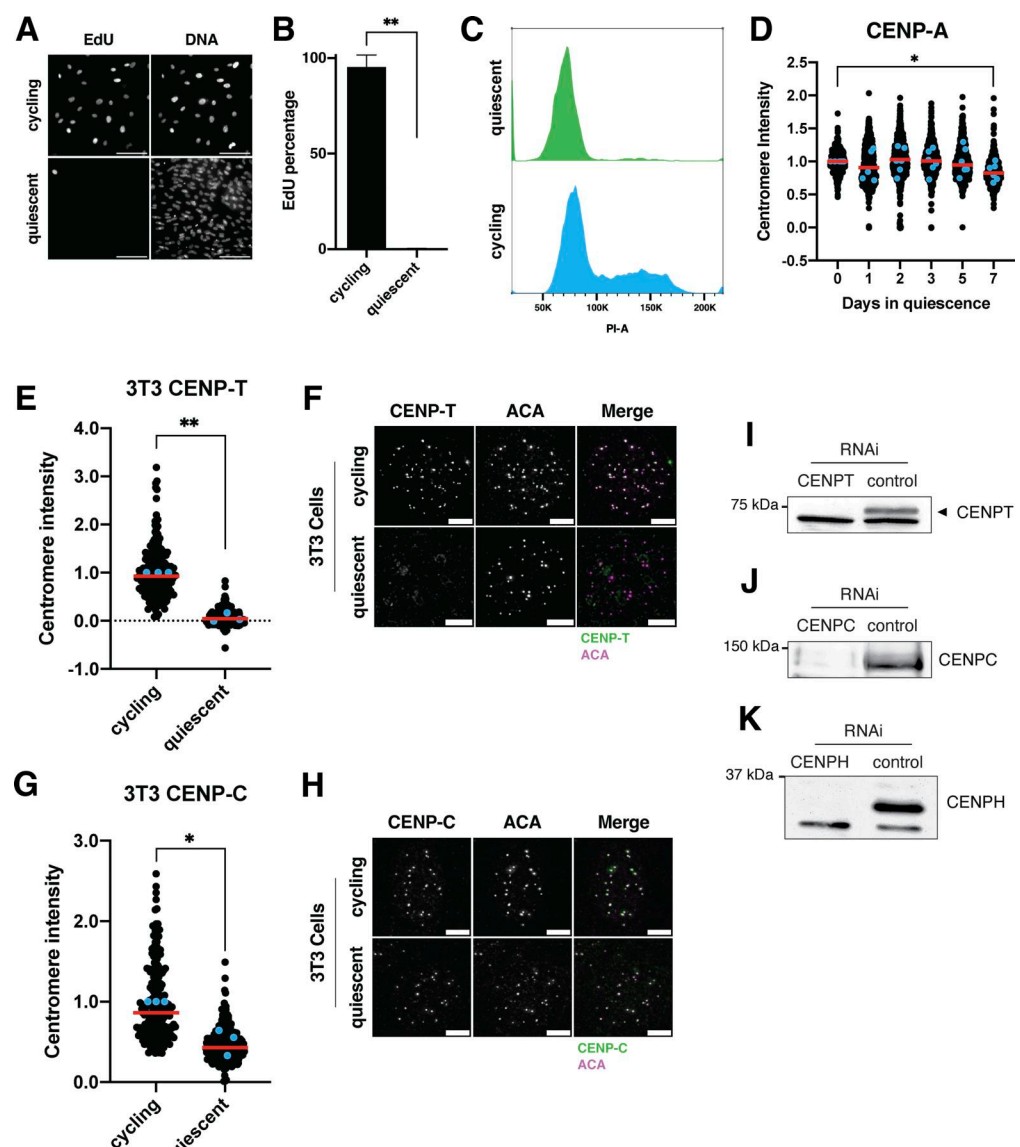

Figure S1. **Controls for quiescence induction and conservation of quiescent centromere behavior in mouse cells. (A)** Representative images of EdU staining of cycling and quiescent cells. Cells were incubated for 48 h in EdU, a nucleotide analog that monitors DNA replication and progression through the cell cycle. Scale bar = 100 µm. **(B)** Graph showing the percentage of EdU-positive cells for the indicated condition. Cells were incubated for 48 h in EdU. Bars represent mean ± SD of three replicates. Mean of cycling is 95.12; mean of quiescence is 0.2. ** represents P = 0.0015, using unpaired $t$ test with Welch's correction. **(C)** Histogram showing the distribution of PI staining for cycling cells (blue) or cells in quiescence for 7 days (green) as measured by flow cytometry. **(D)** Graph showing CENP-A centromere intensity level over time of quiescence entry. Each point indicates the average centromere intensity level for all centromeres of a single cell, adjusted for background. Intensity values were normalized to day 0. Red line represents the median, and blue points represent average of each replicate. Points were from aggregated from six replicates. $n$ = 440, 661, 768, 726, 686, and 555 cells for 0, 1, 2, 3, 5, and 7-day time points, respectively. * represents P < 0.05; P = 0.0297 between cycling and 7-day quiescent, averages were 1 and 0.8429, respectively. P values were calculated using unpaired $t$ test with Welch's correction. **(E)** Graph showing CENP-T centromere intensity levels in cycling and quiescent mouse 3T3 cells. Cells were quiescent for 7 days. Each point indicates the average centromere intensity level for a single cell, adjusted for background. Intensity values were normalized to cycling condition. Red line represents the median, and blue points represent average of each replicate. Points were aggregated from three replicates. $n$ = 206 and 223 cells for cycling and quiescent, respectively. P = 0.0031, using unpaired $t$ test with Welch's correction. **(F)** Representative immunofluorescence images of cycling and 7-day quiescent mouse 3T3 cells. Cells were stained with mouse CENP-T and anti-centromere (ACA) antibodies. Scale bar = 5 µm. **(G)** Graph showing CENP-C centromere intensity levels in cycling and quiescent mouse 3T3 cells. Cells were quiescent for 7 days. Each point indicates the average centromere intensity level for a single cell, adjusted for background. Intensity values were normalized to cycling condition. Red line represents the median, and blue points represent average of each replicate. Points were aggregated from three replicates. $n$ = 195 and 254 cells for cycling and quiescent, respectively. P = 0.0342, using unpaired $t$ test with Welch's correction. **(H)** Representative immunofluorescence images of cycling and 7-day quiescent mouse 3T3 cells. Cells were stained with mouse CENP-C and anti-centromere (ACA) antibodies. Scale bar = 5 µm. **(I)** Western blot of cells treated with 50 nM CENPT or control siRNAs. Blot verifies the banding pattern for CENP-T antibody. CENP-T is the upper band. Blot was incubated in CENP-T antibody. **(J)** Western blot of cells treated with 50 nM CENPC or control siRNAs. Blot verifies the banding pattern for CENP-C antibody. Blot was incubated in CENP-C antibody. **(K)** Western blot of cells treated with 50 nM CENPH or control siRNAs. Blot verifies the banding pattern for CENP-H antibody. CENP-H is the upper band. Blot was incubated in CENP-H antibody. Source data are available for this figure: SourceData FS1.

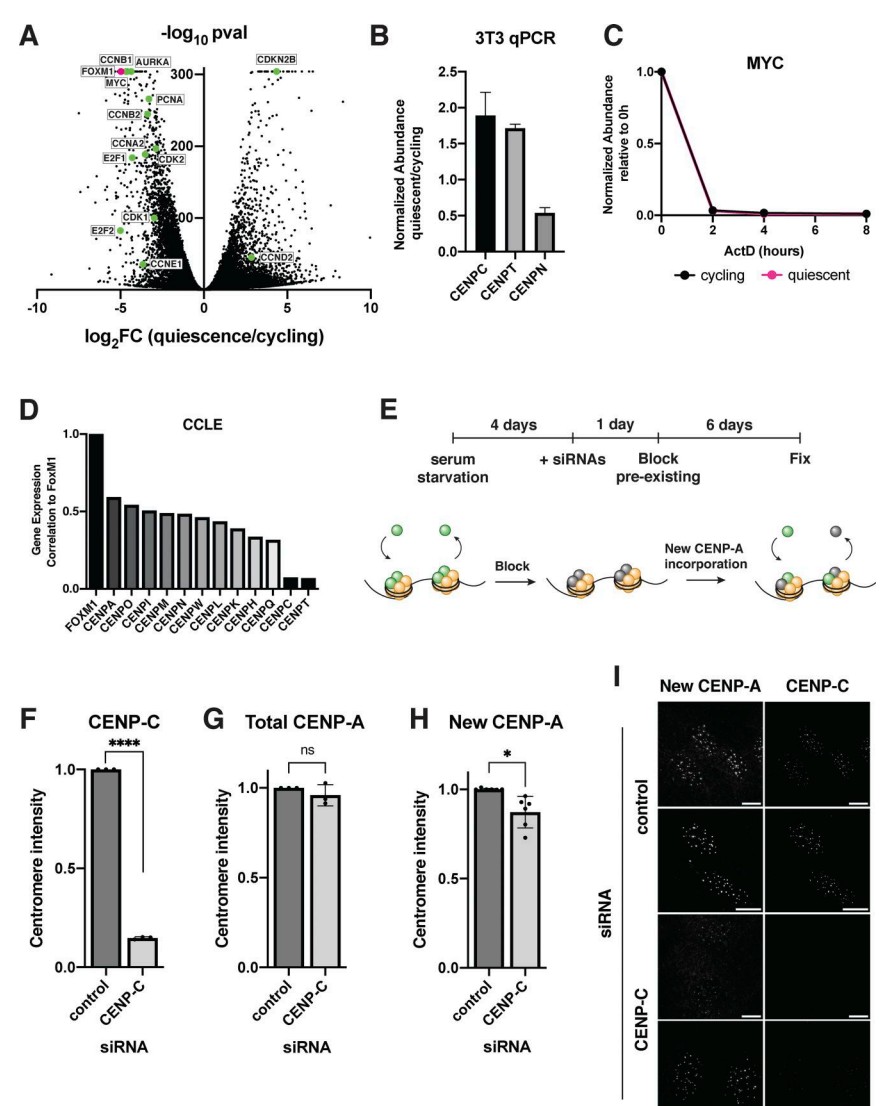

Figure S2. **CENP-C contributes to CENP-A deposition in quiescent cells. (A)** Volcano plot comparing mRNA abundances in quiescent and cycling RPE1 cells as measured by RNA sequencing. Positive controls, including certain cyclins, cyclin-dependent kinases, and other proliferation factors, are highlighted in green. FOXM1 is highlighted in magenta. A P value cutoff was imposed at P = 6.84E–305 for genes with P values of 0. Genes with low read counts (total counts <50) were excluded. **(B)** Graph showing the fold change in mRNA abundance in mouse 3T3 cells between cycling cells and cells in quiescence for 7 days for the indicated centromere component as quantified by qPCR. CT values were normalized to those of GAPDH before comparing quiescent and cycling values. Graph shows at least three biological replicates, with three technical replicates each for each centromere mRNA. Bars represent mean ± SD. **(C)** Graph showing myc mRNA abundance over time after addition of 5 µg/ml actinomycin D in cycling and quiescent cells. Myc mRNA is known to be unstable and have a short half-life. Fold change is calculated for each ActD time point by dividing by untreated (0 h) value for each respective condition (quiescent or cycling). CT values were normalized to those of GAPDH before comparing treated and untreated values. Graph shows three biological replicates, with three technical replicates each. Bars represent mean ± SD. **(D)** Graph showing the correlations in the gene expression profiles of various centromere protein genes and FoxM1 across samples from the Broad Cancer Cell Line Encyclopedia data set (Barretina et al., 2012). Correlations are ordered from largest to smallest and obtained from Thiru et al. (2014). **(E)** Schematic showing experimental design for HaloTag pulse-chase experiments. Unblocked CENP-A is shown in green, blocked CENP-A in gray, and other histones in yellow. More experimental details can be found in the methods section. **(F)** Graph showing CENP-C intensity at the centromeres after 7 days of RNAi treatment. Cells were fixed and stained with CENP-C antibody at the end of the experiment from S2E. Each point indicates the average of centromere intensity level adjusted for background for each replicate. Intensity values were normalized to control. Three replicates were conducted. **** indicates P < 0.0001, using unpaired $t$ test with Welch's correction. $n$ = 486, 432 for control and CENP-C RNAi conditions, respectively **(G)** Graph showing total CENP-A intensity at the centromeres after 7 days of RNAi treatment. Cells were fixed and stained with CENP-A antibody at the end of the experiment from S2E. Each point indicates the average of centromere intensity level adjusted for background for each replicate. Intensity values were normalized to control. Three replicates were conducted. ns = not significant. $n$ = 481, 447 for control and CENP-C RNAi conditions, respectively. P = 0.3528, using unpaired $t$ test with Welch's correction. **(H)** Graph showing new, unblocked CENP-A intensity at the centromeres 6 days after blocking preexisting CENP-A and after 7 days of RNAi treatment. Cells were fixed and stained JF646 at the end of the experiment from S2E. Each point indicates the average of centromere intensity level adjusted for background for each replicate. Intensity values were normalized to control. Six replicates were conducted. * indicates P < 0.05. $n$ = 966, 879 for control and CENP-C RNAi conditions, respectively. P = 0.0161, using unpaired $t$ test with Welch's correction. **(I)** Representative immunofluorescent images showing levels of new unblocked CENP-A and CENP-C at the end of the pulse-chase experiment described in S2E. Cells were treated with either control or CENP-C siRNA. Scale bar = 10 µm.

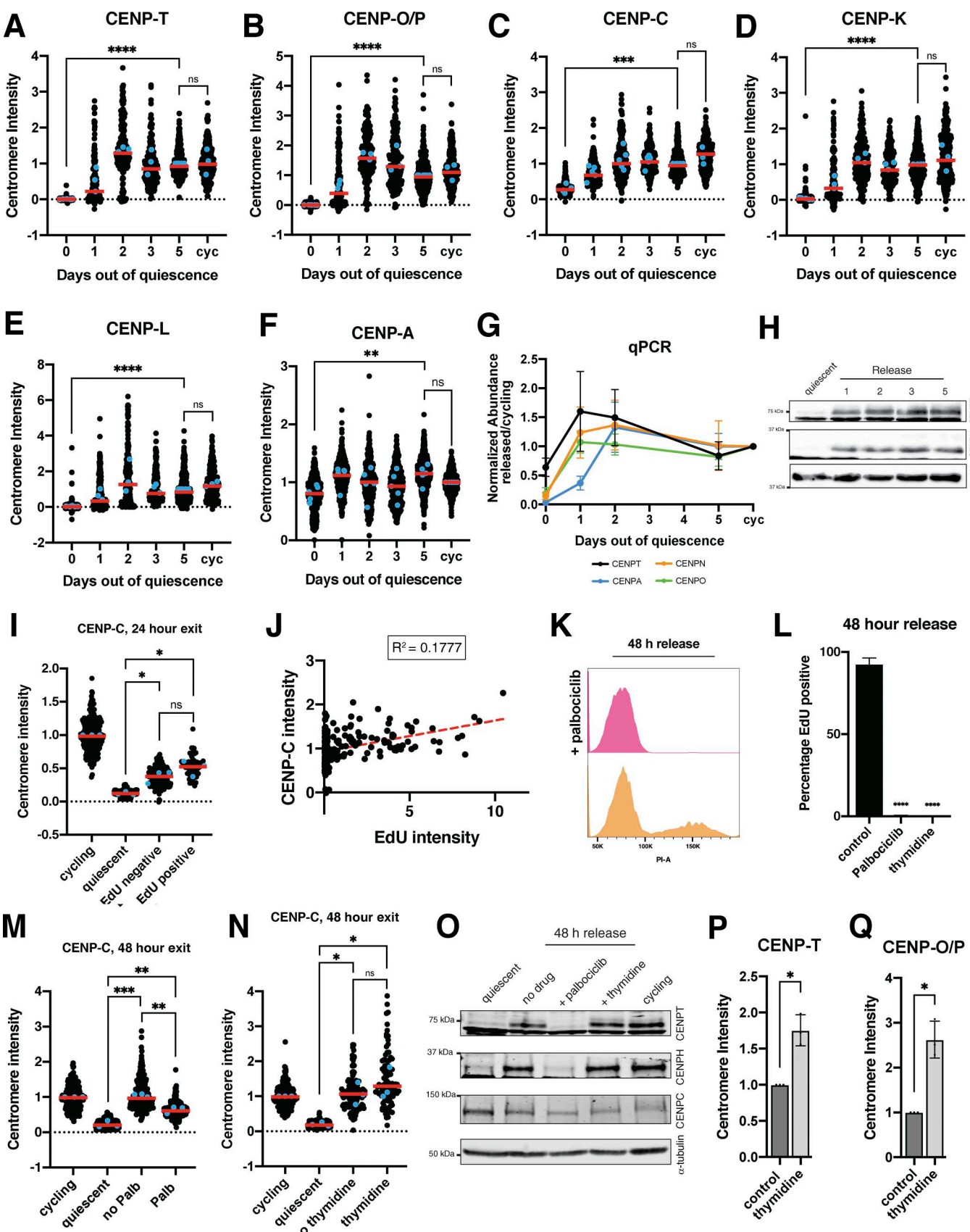

Figure S3.    **The centromere is rapidly reassembled upon cell cycle reentry. (A)** Graph showing CENP-T centromere intensity level over time of quiescence exit. Each point indicates the average centromere intensity level for all centromeres in a single cell, adjusted for background. Intensity values were normalized

to day 5. Red line represents the median, and blue points represent average of each replicate. Points were aggregated from three or more replicates. n = 613, 193, 182, 214, 456, and 336 cells for 0, 1, 2, 3, 5-day, and cycling time points, respectively. P values can be found in Table S2. **(B)** Graph showing CENP-O/P centromere intensity level over time of quiescence exit. Each point indicates the average centromere intensity level for all centromeres in a single cell, adjusted for background. Intensity values were normalized to day 5. Red line represents the median, and blue points represent average of each replicate. Points were aggregated from three or more replicates. n = 618, 200, 184, 218, 416, and 348 cells for 0, 1, 2, 3, 5-day, and cycling time points, respectively. P values can be found in Table S2. **(C)** Graph showing CENP-C centromere intensity level over time of quiescence exit. Each point indicates the average centromere intensity level for all centromeres in a single cell, adjusted for background. Intensity values were normalized to day 5. Red line represents the median, and blue points represent average of each replicate. Points were aggregated from three or more replicates. n = 698, 299, 335, 253, 440, and 346 cells for 0, 1, 2, 3, 5-day, and cycling time points, respectively. *** represents P = 0.0001. P values can be found in Table S2. **(D)** Graph showing CENP-K centromere intensity level over time of quiescence exit. Each point indicates the average centromere intensity level for all centromeres in a single cell, adjusted for background. Intensity values were normalized to day 5. Red line represents the median, and blue points represent average of each replicate. Points were aggregated from three or more replicates. n = 575, 148, 183, 175, 410, and 361 cells for 0, 1, 2, 3, 5-day, and cycling time points, respectively. P values can be found in Table S2. **(E)** Graph showing CENP-L centromere intensity level over time of quiescence exit. Each point indicates the average centromere intensity level for all centromeres in a single cell, adjusted for background. Intensity values were normalized to day 5. Red line represents the median, and blue points represent average of each replicate. Points were aggregated from three or more replicates. n = 554, 281, 213, 210, 462, and 275 cells for 0, 1, 2, 3, 5-day, and cycling time points, respectively. P values can be found in Table S2. **(F)** Graph showing CENP-A centromere intensity level over time of quiescence exit. Each point indicates the average centromere intensity level for all centromeres in a single cell, adjusted for background. Intensity values were normalized to day 5. Red line represents the median, and blue points represent average of each replicate. Points were aggregated from 5 replicates. n = 863, 423, 351, 416, 670, and 527 cells for 0, 1, 2, 3, 5-day, and cycling time points, respectively. P values can be found in Table S2. **(G)** Graph showing mRNA abundance for centromere components over time as cells exit quiescence. Fold change is calculated for each time point by comparing with cycling value. CT values were normalized to those of GAPDH before comparing released and cycling values. Graph shows at least three biological replicates, with three technical replicates each for each centromere mRNA. Bars represent mean ± SD. **(H)** Western blots of cells release from quiescence for the indicated amount of days. Blots were incubated with CENP-T and CENP-H antibodies. β-Actin is used as a loading control. **(I)** Graph showing CENP-C centromere intensity levels for different conditions: cycling, 7 days of quiescence, and either EdU-negative or EdU-positive cells from cells released 24 h from quiescence. Each point indicates the average centromere intensity level for all centromeres in a single cell, adjusted for background. Intensity values were normalized to cycling. Red line represents the median, and blue points represent average of each replicate. Points were aggregated from three replicates. n = 203, 304, 210, and 70 cells for cycling, quiescent, 24 h release EdU negative, and 24 h release EdU positive, respectively. * represents P < 0.05 and ns = not significant. P = 0.220 between EdU negative and EdU positive, P = 0.0259 between EdU positive and quiescent, and P = 0.0632 between EdU negative and quiescent using unpaired t test with Welch's correction. Other pairwise comparisons can be found in Table S2. **(J)** Graph showing CENP-C centromere intensity plotted against EdU intensity for each cell. CENP-C intensity is the average CENP-C centromere intensity level for all centromeres in a cell, adjusted for background. EdU intensity is the mean EdU intensity in the nucleus of the same cell. Points were aggregated from three replicates. Values were normalized within each replicate. $R^2$ = 0.1777. n = 272 cells. **(K)** Histogram showing the distribution of PI staining for cells release 48 h from quiescence without (orange) and with (pink) addition of palbociclib, as measured by flow cytometry **(L)** Graph showing the percentage of EdU-positive cells for the indicated conditions. Cells were released from quiescence into full serum media and incubated for 48 h in EdU with or without palbociclib or thymidine. Bars represent mean ± SD of four replicates for control, three replicates for thymidine, and two replicates for palbociclib. Mean of control is 92.03, mean of palbociclib is 0.3776, and mean of thymidine is 0. **** represents P < 0.0001, using unpaired t test with Welch's correction. **(M)** Graph showing CENP-C centromere intensity for different conditions: cycling, quiescent, 48-h release from quiescence without palbociclib, and 48-h release from quiescence with palbociclib. Each point indicates the average centromere intensity level for all centromeres in a single cell, adjusted for background. Intensity values were normalized to cycling. Red line represents the median, and blue points represent average of each replicate. Points were aggregated from three replicates. n = 242, 346, 260, and 260 for cycling, quiescent, no palbociclib, and palbociclib, respectively. ** represents P < 0.01, ** represents P < 0.001, and ns = not significant. P = 0.0089 between palbociclib and no palbociclib, P = 0.0009 between no palbociclib and quiescent, and P = 0.0056 between palbociclib and quiescent, using unpaired t test with Welch's correction. Other pairwise comparisons can be found in Table S2. **(N)** Graph showing CENP-C centromere intensity for different conditions: cycling, quiescent, 48-h release from quiescence without thymidine, and 48-h release from quiescence with thymidine. Each point indicates the average centromere intensity level for all centromeres in a single cell, adjusted for background. Intensity values were normalized to cycling. Red line represents the median, and blue points represent average of each replicate. Points were aggregated from three replicates. n = 186, 305, 141, and 101 for cycling, quiescent, no thymidine, and thymidine, respectively. * represents P < 0.05; ns = not significant. P = 0.4888 between thymidine and no thymidine, P = 0.0385 between no thymidine and quiescent, and P = 0.0488 between thymidine and quiescent, using unpaired t test with Welch's correction. Other pairwise comparisons can be found in Table S2. **(O)** Western blot of 7-day quiescent cells, cells released from quiescence for 48 h in the presence of no drug, palbociclib, and thymidine, and asynchronous cycling cells. Blot was incubated with CENP-T, CENP-H, and CENP-C antibodies, and α-tubulin is used as a loading control. **(P)** Graph showing CENP-T centromere intensity in cycling cells treated with or without 2 mM thymidine for 48 h. Each point indicates the average of centromere intensity level adjusted for background for each replicate. Intensity values were normalized to control. Points were aggregated from three replicates. n = 894, 337 for control and thymidine, respectively. P = 0.0261 using unpaired t test with Welch's correction. **(Q)** Graph showing CENP-O/P centromere intensity in cycling cells treated with or without 2 mM thymidine for 48 h. Each point indicates the average centromere intensity level for all centromeres in a single cell, adjusted for background. Intensity values were normalized to cycling. Red line represents the median. Points were aggregated from three replicates. n = 422, 197 for control and thymidine, respectively. P = 0.0206 using unpaired t test with Welch's correction. Source data are available for this figure: SourceData FS3.

**Provided online are Table S1 and Table S2. Table S1 shows antibody information and primer sequences used for qPCR. Table S2 shows statistics and P values for all pairwise comparisons between conditions for all figures.**

