## [Peer Review File · The Journal of Cell Biology]

The dynamics of centromere assembly and disassembly during quiescence

Océane Marescal, Kuan-Chung Su, Britannia Moodie, Noah Taylor, and Iain Cheeseman

Corresponding Author(s): Iain Cheeseman, Whitehead Institute for Biomedical Research

Review Timeline:

Submission Date:	2025-09-08
Editorial Decision:	2025-10-28
Revision Received:	2026-01-28
Editorial Decision:	2026-02-25
Revision Received:	2026-03-03

Monitoring Editor: Aaron Straight

Scientific Editor: Andrea Marat

Transaction Report:

DOI: <https://doi.org/10.1083/jcb.202509067>

October 27, 2025

Re: JCB manuscript #202509067

Iain Cheeseman
Whitehead Institute for Biomedical Research

Dear Dr. Cheeseman,

Thank you for submitting your manuscript entitled "The dynamics of centromere assembly and disassembly during quiescence". The manuscript has been evaluated by expert reviewers, whose reports are appended below. Unfortunately, after an assessment of the reviewer feedback, our editorial decision is against publication in JCB.

You will see that while the reviewers overall appreciate the quality of the data provided, there are significant concerns that study does not currently provide a sufficient molecular and conceptual advance for a broad cell biology readership. We agree with their concern that many of the conclusions were predictable from previous work in the field. In particular, the delay of CENP-A assembly until after the first passage through mitosis and the loss of other CCAN components but not CENP-A and CENP-C during quiescence. For the observations that are potentially new as indicated by reviewers 1 and 2 there is little in-depth analysis to understand how centromeres are changing and for the most part just the observation that the CCAN is disassembled. Altogether therefore your study does not seem suitable as a JCB Report, which must represent a highly novel observation.

Therefore, although your manuscript is intriguing, we feel that the points raised by the reviewers are more substantial than can be addressed in a typical revision period. If you wish to expedite publication of the current data, it may be best to pursue publication at another journal.

Given interest in the topic we would be willing to reconsider a significantly expanded study that provides detailed mechanistic insight as well as addresses all concerns regarding the statistical analysis throughout. If you would like to resubmit this work to JCB, please contact the journal office to discuss an appeal of this decision or you may submit an appeal directly through our manuscript submission system. Please note that priority and novelty would be reassessed at resubmission.

If you would like to pursue publication at another journal we encourage you to transfer your study to our not-for-profit open-access sister journal, Life Science Alliance (LSA). We shared your manuscript and the accompanying reviews with LSA Executive Editor, Tim Fessenden, who is interested in these findings. He is pleased to offer consideration of this manuscript at LSA pending the following revisions:

- Address the concerns on statistical analyses throughout this work raised by Reviewer 1 in the manner of your choosing.
- Discuss the observed effects of palbociclib on CENP-C loading as remarked on by Reviewer 1.

We understand that such a revision might need to be re-reviewed by Reviewer 1, in which case Dr. Fessenden will walk them through our transfer process. If interested we encourage you to contact Dr. Fessenden at t.fessenden@life-science-alliance.org to discuss this work and the revisions requested. You may use the link below to immediately transfer your manuscript to LSA. You do not need to revise the manuscript before transferring it to LSA. Once you transfer, you will receive an invitation to revise and resubmit. Again please contact Dr. Fessenden if you have any questions about the LSA journal, the transfer process, or the revisions requested.

Link Not Available

Regardless of how you choose to proceed, we hope that the comments below will prove constructive as your work progresses. We would be happy to discuss the reviewer comments further once you've had a chance to consider the points raised in this letter. You can contact the journal office with any questions at cellbio@rockefeller.edu.

Thank you for your interest in Journal of Cell Biology.

Sincerely,

Aaron Straight
Monitoring Editor

Andrea L. Marat
Deputy Editor

Reviewer #1 (Comments to the Authors (Required)):

This paper by Marescal and colleagues describes the phenomenon of centromere assembly upon exit of cells from cellular quiescence. This work follows up on an earlier report from the authors that describes the maintenance and slow turnover of CENP-A chromatin during quiescence in RPE cells.

The authors find that, upon exit from quiescence, most centromere components assemble in the first S phase following exit from the quiescent state, with the exception of CENP-A that loads in the next G1 cycle after passage through mitosis. This paper is fairly descriptive with little mechanistic insight and while the dynamics of centromere assembly upon re-entry into the cell cycle is novel, the findings are largely expected with little insight into how this is regulated. In addition, experiments are somewhat scattered and poorly controlled, and the discussion is very limited.

In my view the level of execution of the experiments and novelty of the paper is insufficient for what would be expected for JCB. Below, I offer key criticisms and other suggestions to help improve.

Major points

The authors find that CENP-A assembly occurs not immediately upon exit from quiescence but rather following the next mitosis in G1 phase. This would be expected as cells enter quiescence following growth factor depletion after exit from mitosis and entry into G1. Thus, the cells have loaded CENP-A already prior to entry into quiescence. This implies that the permissive window for CENP-A assembly has already passed when cells exit from quiescence. As CENP-A assembly in cycling cells occurs within the first one or two hours of mitotic exit, it is likely that this window will indeed not re-occur upon quiescence exit. While the authors make this observation they offer no discussion on the possible difference in the biochemical states between cells that enter G1 and those that exit quiescence and why they would expect CENP-A to load upon entry into late G1 after re-entry of the cell cycle.

Another reason for the lack of CENP-A assembly immediately upon exit from quiescence may be the lack of CENP-A expression. The authors have performed RNA sequencing and showed that CENP-A levels are reduced. However, analysis of the rate of re-expression of CENP-A and other centromere proteins would offer insight into whether reloading of centromere proteins at the centromere is either linked or uncoupled from their expression.

One major caveat of the analysis of the experiments in this paper is the use of statistics. Several experiments were performed only twice with no statistical analysis possible, making it difficult to discern the relevance of the effects observed. For instance, in Figure 1 or in Figure 2, where no effect of ActD is claimed without any measure of possible differences and stats.

In other cases, such as in Figure 3 and other figures, it is not clear how many replicates of experiments were performed. More problematically, the way that stats are performed here is incorrect. All centromeres measured across different experiments are pooled together and used as input to calculate stats. With hundreds of centromeres measured, any difference, however small will be significant simply because of the large number of datapoints. The number of centromeres is not the measure that is needed to determine significance. What matters is the experimental variation in order to determine whether there is a real effect, which if there are three experiments, then $n=3$. Thus, the average centromere intensity of each experiment should be determined and then the variance between replicate experiments determines whether the observed difference is significant, i.e. likely to be real.

This is a problem throughout the paper and seriously impedes any kind of interpretation of significance. A case in point is the measurement of new CENP-A assembly upon CENP-C depletion, where a four-star significance is claimed over a 4% reduction in total levels and 15% reduction in new CENP-A, but using close to a 1000 datapoints. The authors should re-analyze their data and perform statistics across different replicate experiments.

The mRNA stability measurements in figure S2 are poorly controlled, with no measure of whether ActD effectively inhibits transcription. There are no effects of this treatment on mRNA turnover in any condition, which makes the inclusion of a positive control critical.

The authors find that while most genes encoding centromere proteins are downregulated during quiescence, CENP-C is a notable exception. However, there is no discussion on why this may be the case. Possibly the promoter and sets of transcription factors binding to CENP-C is different from other centromere protein genes that are typically cell cycle regulated genes induced by S-phase-associated transcription factors. Some analysis of this and a discussion on why CENP-C differs from other genes would greatly strengthen the paper.

There is limited discussion on the mechanisms of CENP-A turnover during quiescence. Others have reported on this, such as Saayman et al. (Esashi lab) (PMID: 36702125) who identified elevated DNA damage at centromeres which may be a

mechanism of CENP-A turnover. This work is not cited in this study.

The Palbociclib experiments show an interesting differentiation between the timing of CENP-C assembly and other centromere proteins. While loading of CENP-T appears strictly dependent upon S-phase entry, CENP-C is not. The authors do not specifically comment on this finding. They mention CENP-C is maintained in quiescence but their data indicates that, in addition to maintenance, there is also a further increase in CENP-C upon release from quiescence even in the presence of Palbociclib, indicating very early reloading of some CENP-C, prior to e.g. CENP-T. This observation is not clearly fleshed out.

In Supplemental Figure 1, the authors show that upon entry into quiescence CENP-A levels drop significantly within the first day of quiescence, and are then maintained at a lower level. This rapid drop of CENP-A would give an opportunity to understand this mechanistically. Particularly, the authors have previously suggested that transcription could be responsible for CENP-A turnover (in oocytes). Considering the short time frame in which CENP-A is lost, transcription inhibition might be viable in this time window. Short of this, this observation of rapid loss warrants at least some level of discussion.

Supplemental Figure 2B, qPCR analysis of the CENP-T gene in mouse 3T3 cells shows that expression of these genes is higher in quiescent cells compared to cycling cells, which is inconsistent with the data in RPE cells. This should be explained or discussed.

In Supplemental Figure 3 the authors perform a time course analysis of centromere protein recovery at centromeres following release from quiescence. However, the intensity levels should be compared to cycling levels in order to assess to what extent levels are recovered.

The authors performed elegant live cell experiments demonstrating CENP-A assembly dynamics upon re-entry into the cell cycle. They show that CENP-A behaves differently from other centromere components. However, these are analyzed under different conditions in Figure 3 and Supplemental Figure 3. Specifically, it would be helpful to include the analysis of CENP-A in the experiments performed in Supplemental Figure 3, showing the daily accumulation of centromere proteins. CENP-A is missing from this analysis, and having this alongside the other centromere proteins would be helpful to demonstrate their differences in behaviour.

Minor points

In Supplemental Figure 1 panel I, J & K RNAi experiments are shown for CENP-T, CENP-C, and CENP-H. However, these experiments do not seem to relate to anything else in the paper. Why is this here?

In Figure 1 the time point that is represented in the micrographs should be included.

Similarly, in Supplemental Figure 1, it is not indicated how long cells are treated to reach quiescence in the mouse 3T3 cells.

Also for Figure 2B, qPCR analysis, when is this analysis performed?

The authors stain for CENP-O/P in several of the experiments. Does this mean that the antibody recognises a complex of these two proteins or is it unknown which epitope is detected? This is not explained.

With the analysis of multiple centromere proteins along the time course of exit from quiescence it would be interesting to determine if there is any order of recovery.

The experiments demonstrating that CENP-C is required to maintain CENP-A levels during quiescence is consistent with the authors' earlier work showing that Mis18 β and HJURP are also required. This should be mentioned and cited to put this finding into context.

Reviewer #2 (Comments to the Authors (Required)):

Centromeres are critical structures during cell division. However, we know very little about these structures when cells exist the cell cycle and stop dividing. The manuscript from Marescal and colleagues defines the changes to the centromere as cells enter quiescence. They find that components of the CCAN are drastically and quickly reduced as cells enter quiescence. An exception is CENP-C, which persists at centromeres after quiescence, as does CENP-A (demonstrated previously). As quiescence is reversible, the authors go on to determine how the centromeres respond to re-entry into the cell cycle. They find that the CCAN proteins are recircuited to centromeres in S-phase, and that CENP-A nucleosomes are assembled in the following G1. Overall, the data clearly demonstrate the dynamics of CCAN in response to cell cycle exit. The experiments are conducted well and are informative. However, the work is primarily observational and does not give any mechanistic insight into the phenomena that are describing. Additional experiments that address the mechanisms that control the disassembly and re-recruitment of the CCAN

would significantly add to the manuscript.

1. The experiments show the reduction in the CCAN correlates with decreased gene expression. Is the loss of CCAN proteins driven by the reduction in transcript level, or is the centromere unable to recruit CCAN proteins during quiescence? Does a dox-inducible CENP-L or CENP-T get recruited to centromere or are they refractory to CCAN assembly during quiescence.
2. If the process is driven by protein availability, then what is the pathway that is controlling expression. Is this the FoxM1/Myb12 pathway that has been implicated previously by the group? How does expression change in these proteins with quiescence?
3. In the context of the Palbocilic and thymidine experiments when cells are reentering the cell cycle (fig. 4), is the lack of re-recruitment prior to S-phase due to the lack of CCAN gene expression until cells have committed to S-phase or is protein recruitment limited to S-phase.

Reviewer #3 (Comments to the Authors (Required)):

This study investigates centromere dynamics in quiescent, non-dividing human cells. Remarkably, entry into and exit from quiescence provides a unique system in which to examine the disassembly and reassembly of the centromere.

A major finding is that upon entry into quiescence, in addition to low levels of CENP-A, some residual CENP-C is maintained at centromeres. This low level of CENP-C functions in the low rate of de novo CENP-A assembly which maintains the centromere during the quiescent state. In contrast, other CCAN components are rapidly lost from the centromere, and are downregulated at the transcriptional level. The authors also show that upon exit from quiescence, the CCAN reassembles in S phase, whereas CENP-A is replenished after exit from mitosis, in the next G1 phase. Remarkably, the amount of CENP-A loaded at the centromere is restored to the typical level of cycling cells in this single loading event. These conclusions are very well supported by the data presented. The study can essentially be published as is, but I have a two very minor points that could be interesting to discuss.

- Upon exit from quiescence, CENP-C can be assembled prior to entry into S phase (Figure S3G, S3K), even earlier than other CCAN components. Given that CENP-C binds directly to CENP-A nucleosomes, how does efficient CENP-C recruitment occur when the CENP-A level is still very low?
- It is also curious that new CENP-A does not load in the first G1 phase. Presumably, the deposition machinery is not recruited/available. What is known about PKL1/CDK1 activity upon exit from quiescence?

Minor points:

- For Figure 1, which time point is shown in images 1B, 1D, 1F etc?
- The image shown in S2G looks like more than a 15% difference? Is it representative?

We thank the reviewers for their constructive comments and suggestions. For the revised manuscript, we have made changes to the text and figures in response to these suggestions. In particular, we have added multiple new experiments that more deeply analyze the mechanism behind centromere protein loss during quiescence and the behavior of centromere protein reacquisition upon the return to growth. These include:

1. **Determining the underlying factors preventing CENP-A re-deposition during the G1 that follows quiescence exit.** We previously found that CENP-A deposition does not occur during the G1 immediately following quiescence exit, but instead occurs in the next G1 after cells complete their first mitosis. We have now assessed both PLK1 localization and CDK1 activity following exit from quiescence - two important regulators of CENP-A deposition. CDK1 inactivation upon mitotic exit is a key event that allows CENP-A deposition to occur. We find that quiescent cells exiting quiescence into G1 do not have detectable levels of cyclin B1, indicative of a lack of CDK1/cyclin B activity. Despite this absence of cyclin B, these cells do not deposit CENP-A. However, based on immunofluorescence, we now find that PLK1, whose activity is required for CENP-A deposition, does not localize to centromeres in quiescence-released G1 cells (Figure 3H, I). Based on these data, we propose that the ability of cells to distinguish a normal G1 from a G1-like phase following quiescence release depends on the absence of PLK1 preventing CENP-A deposition. This represents a novel model for the control of CENP-A deposition and explains the molecular logic behind the dual regulatory control of CENP-A deposition. In addition, we have now conducted qPCR for CENP-A for a time course of cells exiting quiescence. We find that CENP-A expression remains below half of that observed in cycling cells immediately after release but increases dramatically after two days. This lower amount of CENP-A expression recovery within 24 hours may also contribute to the lack of CENP-A re-assembly during the initial G1 following exit.
2. **Analyzing the basis for centromere disassembly in quiescent cells.** We previously demonstrated that most centromere proteins are no longer expressed in quiescent cells. For this revised version, we have now tested whether the loss of CCAN localization is due to this decreased expression or whether other factors are present that prevent CCAN protein recruitment during quiescence even upon centromere protein overexpression. To test this, we generated a cell line ectopically expressing GFP-CENP-N, which is a direct binding partner for CENP-A. Despite complete loss of endogenous CENP-L/N in quiescent cells, ectopically expressed GFP-CENP-N localized to centromeres in quiescent cells. Thus, reduced gene expression for CCAN proteins is a primary determinant of their inability to localize to quiescent centromeres.
3. **Analyzing the effects of protein availability on the timing of centromere re-deposition upon return to growth.** In the prior version of the paper, we observed that CENP-T and CENP-O/P relocalize to centromeres in S phase during the return to growth from quiescence, whereas CENP-C re-localizes earlier. To understand this differential behavior, we have now probed for CCAN proteins by western blot during 48 hours of quiescence release in the presence of Palbociclib (G1 arrest) and Thymidine (S phase arrest). We find that cells exiting quiescence and arrested in G1 have not yet regained CENP-T and CENP-H protein levels, but retain high CENP-C levels. Indeed, CENP-T and CENP-H protein levels are not regained until S phase. Thus, the earlier re-deposition of CENP-C and the timing of CENP-T and CENP-H redeposition during S phase may rely on the availability of these proteins upon quiescence release.
4. **Implicating FoxM1 as a transcription factor regulating centromere protein gene expression during quiescence.** We previously found that all centromere proteins are downregulated at the gene expression level during quiescence, with the exception of CENP-C, whose expression increases in quiescent cells. To identify possible regulatory

pathways for the coordinated downregulation of centromere proteins in quiescent cells and explain the unique behavior of CENP-C expression, we have now reanalyzed available gene expression and ChIP-seq data. We now find that there is a strong correlation between the expression of the transcription factor FoxM1 and most centromere genes, but not CENP-C. Similarly, we observe a relative enrichment of FoxM1 binding to promoters of most centromere protein genes, but not to that of CENP-C. We also find that FoxM1 is highly downregulated (~30-fold) in quiescent cells according to our RNA-seq data. Thus, FoxM1 may regulate the transcription of other centromere proteins, with the exception of CENP-C, and the downregulation of FOXM1 in quiescent cells could regulate the general decrease in centromere protein expression, while not affecting CENP-C.

5. **Determining CCAN gene expression upon return to growth.** For this revised version of the manuscript, we have now performed qPCR across a time course as cells exit quiescence and reenter the cell cycle to analyze the rates of re-expression for CENPA, CENPT, CENPN, and CENPO. We observe increased expression above the levels observed in cycling cells for all centromere components at early time points following the exit from quiescence. Expression levels subsequently stabilize following an extended return to growth.
6. **Improving the calculation of statistics and increasing the number of replicates.** We have now calculated all statistics and p-values based on the averages of replicates and have included extra replicates for entry and exit from quiescence time courses and for the ActD mRNA degradation experiment.

Below we include detailed point-by-point responses to each reviewer comment.

Reviewer #1

1. The authors find that CENP-A assembly occurs not immediately upon exit from quiescence but rather following the next mitosis in G1 phase. This would be expected as cells enter quiescence following growth factor depletion after exit from mitosis and entry into G1. Thus, the cells have loaded CENP-A already prior to entry into quiescence. This implies that the permissive window for CENP-A assembly has already passed when cells exit from quiescence. As CENP-A assembly in cycling cells occurs within the first one or two hours of mitotic exit, it is likely that this window will indeed not re-occur upon quiescence exit. While the authors make this observation they offer no discussion on the possible difference in the biochemical states between cells that enter G1 and those that exit quiescence and why they would expect CENP-A to load upon entry into late G1 after re-entry of the cell cycle. Another reason for the lack of CENP-A assembly immediately upon exit from quiescence may be the lack of CENP-A expression. The authors have performed RNA sequencing and showed that CENP-A levels are reduced. However, analysis of the rate of re-expression of CENP-A and other centromere proteins would offer insight into whether reloading of centromere proteins at the centromere is either linked or uncoupled from their expression.

Thank you for the suggestion. We have now performed qPCR across a time course as cells exit quiescence and reenter the cell cycle to analyze the rates of re-expression for CENPA, CENPT, CENPN, and CENPO (Figure S3G). Interestingly, we observe increased expression above the levels observed in cycling cells for all centromere components at early time points following the exit from quiescence. Expression levels subsequently stabilize following an extended return to growth. In the case of CENPA, expression remains below half of that observed in cycling cells immediately after release but increases dramatically after two days. This lower amount of CENP-A expression recovery within 24 hours may contribute to the lack of CENP-A re-assembly during the initial G1 following quiescence exit, but we note that it is present at this stage.

In addition, we have also now assessed both PLK1 localization and CDK1 activity following exit from quiescence. In cycling cells, CDK1 inactivation upon mitotic exit is a key event that allows CENP-A deposition to occur. Prior work found that CDK inhibition can induce CENP-A deposition even outside of its normal deposition window. We found that quiescent cells exiting quiescence into G1 do not have detectable levels of cyclin B1 (Figure 3G) indicative of a lack of CDK1/cyclin B activity, such that this should create a permissive window to deposit CENP-A. However, despite the absence of cyclin B, these cells do not deposit CENP-A.

In addition to reduced CDK activity, the other key regulatory step that initiates CENP-A deposition during a normal G1 is positive activity from Plk1. Plk1 localizes robustly to centromeres during G1 in cycling cells. However, based on immunofluorescence analysis, we now find that PLK1 does not localize to centromeres in quiescence-released G1 cells (Figure 3H, I). Given these data, we propose that the ability of cells to distinguish a normal G1 from a G1-like phase following the release from quiescence depends on the absence of PLK1, preventing CENP-A deposition.

- One major caveat of the analysis of the experiments in this paper is the use of statistics. Several experiments were performed only twice with no statistical analysis possible, making it difficult to discern the relevance of the effects observed. For instance, in Figure 1 or in Figure 2, where no effect of ActD is claimed without any measure of possible differences and stats.

In other cases, such as in Figure 3 and other figures, it is not clear how many replicates of experiments were performed. More problematically, the way that stats are performed here is incorrect. All centromeres measured across different experiments are pooled together and used as input to calculate stats. With hundreds of centromeres measured, any difference, however small will be significant simply because of the large number of datapoints. The number of centromeres is not the measure that is needed to determine significance. What matters is the experimental variation in order to determine whether there is a real effect, which if there are three experiments, then $n=3$. Thus, the average centromere intensity of each experiment should be determined and then the variance between replicate experiments determines whether the observed difference is significant, i.e. likely to be real. This is a problem throughout the paper and seriously impedes any kind of interpretation of significance. A case in point is the measurement of new CENP-A assembly upon CENP-C

depletion, where a four-star significance is claimed over a 4% reduction in total levels and 15% reduction in new CENP-A, but using close to a 1000 datapoints. The authors should re-analyze their data and perform statistics across different replicate experiments.

Thank you for this point. We have now added a third replicate for the immunofluorescence time courses for quiescence entry and exit (Figure 1 and Supplementary Figure 3) and for the ActD experiments (Figure 2). We also now specify the number of replicates in each of our figures. We agree with the reviewer that significance should be determined based on averages of the replicate experiments. However, we also believe that showing the spread of individual data points is crucial for showing the range of behaviors across different cells. For example, in our release experiments, we often observe a small population of cells that have not yet recovered centromere protein localization, likely representing cells that have not exited quiescence. This observation is lost if we were only to display the average values. Therefore, for some graphs, we have chosen to keep the underlying datapoints, while also adding points representing the average of each replicate. All statistics were calculated from these replicate averages. For other experiments (such as measurements of new CENP-A assembly), we followed the suggestion from this reviewer and only show the experimental replicate averages, with the correspondingly calculated statistics. For some graphs with multiple points of comparison, we only display the statistics for the crucial comparisons for clarity. However, we have now also included the calculated statistics for all comparisons in Supplementary Table 2.

Below, we show examples of graphs where the replicate averages are displayed in blue and the underlying individual data points in black, with statistics calculated from the blue points. The importance of including black data points is highlighted by the 2-day release point for CENP-T and by the thymidine condition for CENP-O/P.

3. The mRNA stability measurements in figure S2 are poorly controlled, with no measure of whether ActD effectively inhibits transcription. There are no effects of this treatment on mRNA turnover in any condition, which makes the inclusion of a positive control critical.

The mRNAs for all CENP genes tested were destabilized upon addition of ActD in both quiescent and cycling cells, with similar rates of degradation over time between both conditions. To provide a comparison as suggested by this reviewer, we have now added a positive control, *MYC*, whose RNA is known to undergo rapid degradation (Dani *et al.* 1984; PMID: 6594679).

4. The authors find that while most genes encoding centromere proteins are downregulated during quiescence, CENP-C is a notable exception. However, there is no discussion on why this may be the case. Possibly the promoter and sets of transcription factors binding to CENP-C is different from other centromere protein genes that are typically cell cycle regulated genes induced by S-phase-associated transcription factors. Some analysis of this and a discussion on why CENP-C differs from other genes would greatly strengthen the paper.

Thank you for the comment. We agree that the unique upregulation of CENP-C during quiescence and the potential promoter and transcription factors involved are interesting questions. To address this, we have now analyzed available expression data and ChIP-seq data sets (Figure 2K, L, Figure S2D). One of the key factors that controls the expression of cell cycle-regulated genes is the transcription factor FOXM1. We find that there is a clear correlation between the expression of FOXM1 and that of most centromere genes (Figure 2K, Figure S2D). In contrast, there is no such a correlation for CENP-C. In addition, there is a clear enrichment for FOXM1 binding to promoters based on ChIP-seq data for most centromere proteins, but not for CENP-C (Figure S2D). Based on this analysis, we hypothesize that FOXM1 (together with additional potential transcription factors) regulates the transcription of other centromere proteins, but not CENP-C. Importantly, we find that FOXM1 is highly downregulated in quiescent cells in our RNA-seq data (~30-fold) (Figure S2A). Thus, the downregulation of FOXM1 transcription factor in quiescent cells could regulate the decrease in centromere protein expression, while not affecting CENP-C. We have now included these observations in the Results and Discussion section.

- There is limited discussion on the mechanisms of CENP-A turnover during quiescence. Others have reported on this, such as Saayman et al. (Esashi lab) (PMID: 36702125) who identified elevated DNA damage at centromeres which may be a mechanism of CENP-A turnover. This work is not cited in this study.

We did not investigate CENP-A turnover in quiescence in this study, as this was the focus of a previous paper from our lab (Swartz et al. 2019) (PMID: 31422918). However, we appreciate the reviewer highlighting this important additional reference. We have now cited Saayman et al. in our introduction.

- The Palbociclib experiments show an interesting differentiation between the timing of CENP-C assembly and other centromere proteins. While loading of CENP-T appears strictly dependent upon S-phase entry, CENP-C is not. The authors do not specifically comment on this finding. They mention CENP-C is maintained in quiescence but their data indicates that, in addition to maintenance, there is also a further increase in CENP-C upon release from quiescence even in the presence of Palbociclib, indicating very early reloading of some CENP-C, prior to e.g. CENP-T. This observation is not clearly fleshed out.

We agree with the reviewer that this is a fascinating observation. One explanation for this could be protein availability. We have now conducted Western Blots in cells exiting quiescence in the presence of the CDK4/6 inhibitor Palbociclib or the DNA replication inhibitor thymidine (Figure S30). We find that CENP-T protein is not present in cells treated with Palbociclib (which arrests cells prior to S phase), whereas CENP-C protein is present. Thus, protein levels could be one factor that contributes to these different behaviors. However, the different behaviors of these proteins during quiescence, in cycling cells, and during the exit from quiescence suggests that there are also additional factors that contribute to the behavior for these proteins. We now further discuss this observation in the section.

distinct loading proteins. We this Discussion

7. In Supplemental Figure 1, the authors show that upon entry into quiescence CENP-A levels drop significantly within the first day of quiescence, and are then maintained at a lower level. This rapid drop of CENP-A would give an opportunity to understand this mechanistically. Particularly, the authors have previously suggested that transcription could be responsible for CENP-A turnover (in oocytes). Considering the short time frame in which CENP-A is lost, transcription inhibition might be viable in this time window. Short of this, this observation of rapid loss warrants at least some level of discussion.

Thank you for the suggestion. We have now repeated this CENP-A time course and find that, although final CENP-A levels are lower in quiescent cells, this rapid drop does not occur significantly in the first few days of quiescence induction.

8. Supplemental Figure 2B, qPCR analysis of the CENP-T gene in mouse 3T3 cells shows that expression of these genes is higher in quiescent cells compared to cycling cells, which is inconsistent with the data in RPE cells. This should be explained or discussed.

Similar to the reviewer, we found it interesting that CENP-T expression is high in quiescent 3T3 cells. In human RPE1 cells, although CENP-T expression decreases, this decrease is not as drastic as that of other centromere proteins (~25% decrease as opposed to a 70-75% decrease for CENP-N, CENP-I, and CENP-O; Figure 2B). Therefore, CENP-T expression behavior is somewhat similar between mouse 3T3 and human RPE1 cells. As CENP-T does not localize to quiescent centromeres in either 3T3 or RPE1 cells, despite its continued expression, it is possible that post-translational mechanisms could regulate CENP-T localization or levels in both organisms. We now have added comments to this in the Discussion section.

9. In Supplemental Figure 3 the authors perform a time course analysis of centromere protein recovery at centromeres following release from quiescence. However, the intensity levels should be compared to cycling levels in order to assess to what extent levels are recovered.

Thank you for this suggestion. We have now included a comparison to cycling cells for our quiescence release time course experiments (Figure S3A-F). Cells released from quiescence for 5 days recover centromere protein levels similar to those found in cycling cells.

10. The authors performed elegant live cell experiments demonstrating CENP-A assembly dynamics upon re-entry into the cell cycle. They show that CENP-A behaves differently from other centromere components. However, these are analyzed under different conditions in Figure 3 and Supplemental Figure 3. Specifically, it would be helpful to include the analysis of CENP-A in the experiments performed in Supplemental Figure 3, showing the daily accumulation of centromere proteins. CENP-A is missing from this analysis, and having this alongside the other centromere proteins would be helpful to demonstrate their differences in behaviour

Thank you for this suggestion. We have now conducted a time course experiment analyzing CENP-A following the release from quiescence (Figure S3F).

11. In Supplemental Figure 1 panel I, J & K RNAi experiments are shown for CENP-T, CENP-C, and CENP-H. However, these experiments do not seem to relate to anything else in the paper. Why is this here?

The antibodies used for western blots in this study show several non-specific bands. We therefore conducted these siRNA experiments to verify which band corresponds to a given centromere protein. To clarify this, we have changed the wording in the figure legend from “Blot shows the banding pattern...” to “Blot verifies the banding pattern...” We have now also mentioned this point in the Methods section.

12. In Figure 1 the time point that is represented in the micrographs should be included. Similarly, in Supplemental Figure 1, it is not indicated how long cells are treated to reach quiescence in the mouse 3T3 cells.
Also for Figure 2B, qPCR analysis, when is this analysis performed?

For these experiments, cells were in quiescence for 7 days. We have now added this information to the figure legends. For the qPCR in Figure 2B, cells were also in quiescence for 7 days, as was indicated in the figure legend. For further clarification, we have edited the Methods section to indicate that “Quiescent cells were serum starved for 7 days unless otherwise noted.”

13. The authors stain for CENP-O/P in several of the experiments. Does this mean that the antibody recognises a complex of these two proteins or is it unknown which epitope is detected? This is not explained.

This antibody was generated in a previous study (McKinley et al., 2015), and was generated against and recognizes both full-length CENP-O and CENP-P (it is much easier to purify these proteins as a dimer). We have added clarification to the antibody in our supplementary antibody file (Supplementary Table 1).

14. With the analysis of multiple centromere proteins along the time course of exit from quiescence it would be interesting to determine if there is any order of recovery.

We agree with the reviewer that this is a fascinating question. From our current data analyzing the release from quiescence, all centromere proteins tested showed similar recovery patterns within 1, 2, 3, and 5 days of exit, making it difficult to determine an order of recovery. This analysis would require shorter time points, perhaps 12, 16, 20-hour time points, which we believe is beyond the scope of the current study.

15. The experiments demonstrating that CENP-C is required to maintain CENP-A levels during quiescence is consistent with the authors' earlier work showing that Mis18 β and HJURP are also required. This should be mentioned and cited to put this finding into context.

Thank you for this point. We have now mentioned this point and cited this in the text.

Reviewer #2

1. The experiments show the reduction in the CCAN correlates with decreased gene expression. Is the loss of CCAN proteins driven by the reduction in transcript level, or is the centromere unable to recruit CCAN proteins during quiescence? Does a dox-inducible CENP-L or CENP-T get recruited to centromere or are they refractory to CCAN assembly during quiescence.

Thank you for this question and the experiment suggestion. To test this, we generated a cell line ectopically expressing the CENP-A binding partner GFP-CENP-N and induced these cells to enter quiescence. Although CENP-L/N is usually completely lost from centromeres in quiescent cells, cells ectopically expressing GFP-CENP-N retained GFP-CENP-N centromere localization (Figure 2I, J). This localization was slightly reduced when compared to cycling cells, which could be due to the lack of the CENP-N binding partner, CENP-L, in quiescent cells or to the reduced levels of CENP-C. These results suggest that the loss of at least some CCAN protein is not due to the fact that the centromere is unable to recruit CCAN protein during quiescence, but rather because there is not CCAN proteins available due to reduced gene expression.

2. If the process is driven by protein availability, then what is the pathway that is controlling expression. Is this the FoxM1/Myb12 pathway that has been implicated previously by the group? How does expression change in these proteins with quiescence?

Thank you for the excellent suggestion. Our data showed that cells entering quiescence downregulate the expression of all centromere components, with the exception of CENP-C, which was uniquely upregulated. To test whether the FOXM1 transcription factor could be involved in this unique expression pattern in quiescent cells, we re-analyzed publicly available data for FOXM1. These data show that there is a strong correlation between the expression of FOXM1 and most centromere genes, but not CENP-C (Figure 2K, Figure

S3D). In addition, the relative enrichment of FOXM1 binding to promoters of centromere protein genes based on ChIP-seq data is lower for CENP-C than for other centromere proteins (Figure 2L). From this analysis, we hypothesize that FOXM1 may regulate the transcription of other centromere proteins, but not that of CENP-C.

Importantly, following this reviewer's suggestion, we also find that FOXM1 is highly downregulated in quiescent cells in our RNA-seq data (~30-fold) (Figure S2A). We have now highlighted this point in the text and the figures. Thus, the downregulation of FOXM1 transcription factor in quiescent cells could regulate the general decrease in centromere protein expression, while not affecting CENP-C. We have now included these observations in the Results and Discussion section.

3. In the context of the Palbociclib and thymidine experiments when cells are reentering the cell cycle (fig. 4), is the lack of re-recruitment prior to S-phase due to the lack of CCAN gene expression until cells have committed to S-phase or is protein recruitment limited to S-phase.

Thank you for the helpful suggestion. We have now assayed the protein levels of CENP-T, CENP-H, and CENP-C by western blot in cells exiting quiescence for 48 hours either with no drug, or using Palbociclib or thymidine treatment (Figure S3O). We found that CENP-T and CENP-H protein was not recovered in cells treated with Palbociclib, whereas those treated with thymidine had normal centromere protein levels. Although we do not have an

antibody for CENP-O or CENP-P that works for western blotting, we suspect that these proteins would show the same behavior. Therefore, we conclude that the lack of localization prior to S-phase seen in Figure 4 is controlled at least in part by the absence of these CCAN proteins, rather than their lack of recruitment. CENP-C protein, on the other hand, is present both in quiescent cells and in cells released from quiescence for 48 hours in all conditions, which could help explain the differential re-localization behavior of CENP-C in Palbociclib-treated released cells (Figure S3M).

Reviewer #3

1. Upon exit from quiescence, CENP-C can be assembled prior to entry into S phase (Figure S3G, S3K), even earlier than other CCAN components. Given that CENP-C binds directly to CENP-A nucleosomes, how does efficient CENP-C recruitment occur when the CENP-A level is still very low?

Thank you for the great question. Although CENP-A levels are still low at that time point, CENP-C recruitment could still occur in non-stoichiometric ratios during quiescence exit. CENP-C contains two separate CENP-A binding domains, including the Central Domain and the CENP-C motif at the C-terminus. In addition, CENP-C has been shown to dimerize, which could contribute to increased CENP-C recruitment despite low CENP-A levels. We now comment on this point in the Discussion.

2. It is also curious that new CENP-A does not load in the first G1 phase. Presumably, the deposition machinery is not recruited/available. What is known about PLK1/CDK1 activity upon exit from quiescence?

Thank you for this excellent suggestion. We have now conducted further experiments that analyze PLK1 localization and CDK1 activity in cells released from quiescence. We find that quiescent cells exiting quiescence into G1 have low levels of cyclin B1 (Figure 3G) indicative of low CDK1/cyclin B activity – a state that should be permissive for CENP-A deposition. Importantly, using immunofluorescence, we now show that PLK1 does not localize to the centromeres in G1 cells released from quiescence (Figure 3H, I). Therefore, although cells exiting quiescence lack CDK1 activity, these cells would not be able to deposit CENP-A at this time, as they lack proper PLK1 localization. This provides an

elegant mechanism to distinguish between these different cell cycle states to control CENP-A differentially across different conditions.

3. For Figure 1, which time point is shown in images 1B, 1D, 1F etc?

Cells were in quiescence for 7 days. We have now added this information to the figure legend.

4. The image shown in S2G looks like more than a 15% difference? Is it representative?

Thank you for the comment. We have now included a second image.

February 25, 2026

RE: JCB Manuscript #202509067R-A

Iain Cheeseman
Whitehead Institute for Biomedical Research

Dear Dr. Cheeseman:

Thank you for submitting your revised manuscript entitled "The dynamics of centromere assembly and disassembly during quiescence". We would be happy to publish your paper in JCB pending final revisions necessary to meet our formatting guidelines (see details below). In your final revision, please be sure to add and discuss the reference noted by reviewer #2.

A. MANUSCRIPT ORGANIZATION AND FORMATTING:

- 1) Text limits: Character count for Reports is < 20,000, not including spaces. Count includes abstract, introduction, combined results and discussion, and acknowledgments. Count does not include title page, figure legends, materials and methods, references, tables, or supplemental legends.
- 2) Figures limits: Reports may have up to 5 main text figures.
- 3) Figure formatting: Scale bars must be present on all microscopy images, including inset magnifications. Molecular weight or nucleic acid size markers must be included on all gel electrophoresis. Aspect ratios of images may not be altered.
- 4) Statistical analysis: Error bars on graphic representations of numerical data must be clearly described in the figure legend. The number of independent data points (n) represented in a graph must be indicated in the legend. Statistical methods should be explained in full in the materials and methods. For figures presenting pooled data the statistical measure should be defined in the figure legends. Please also be sure to indicate the statistical tests used in each of your experiments (either in the figure legend itself or in a separate methods section) as well as the parameters of the test (for example, if you ran a t-test, please indicate if it was one- or two-sided, etc.). Also, if you used parametric tests, please indicate if the data distribution was tested for normality (and if so, how). If not, you must state something to the effect that "Data distribution was assumed to be normal but this was not formally tested."
- 5) Abstract and title: The abstract should be no longer than 160 words and should communicate the significance of the paper for a general audience. The title should be less than 100 characters including spaces. Make the title concise but accessible to a general readership.
- 6) Materials and methods: Should be comprehensive and not simply reference a previous publication for details on how an experiment was performed. * Please provide full descriptions in the text for readers who may not have access to referenced manuscripts.
- 7) All antibodies, cell lines, animals, and tools used in the manuscript should be described in full, including accession numbers for materials available in a public repository such as the Resource Identification Portal. Please be sure to provide the sequences for all of your primers/oligos and RNAi constructs in the materials and methods. You must also indicate in the methods the source, species, and catalog numbers (where appropriate) for all of your antibodies. Please also indicate the acquisition and quantification methods for immunoblotting/western blots.
- 8) Microscope image acquisition: The following information must be provided about the acquisition and processing of images:
 - a. Make and model of microscope
 - b. Type, magnification, and numerical aperture of the objective lenses
 - c. Temperature
 - d. Imaging medium
 - e. Fluorochromes
 - f. Camera make and model
 - g. Acquisition software
 - h. Any software used for image processing subsequent to data acquisition. Please include details and types of operations involved (e.g., type of deconvolution, 3D reconstitutions, surface or volume rendering, gamma adjustments, etc.).

10) Supplemental materials: There are strict limits on the allowable amount of supplemental data. Reports may have up to 3 supplemental figures. Please also note that tables, like figures, should be provided as individual, editable files. A summary of all supplemental material should appear at the end of the Materials and methods section.

13) ORCID IDs: ORCID IDs are unique identifiers allowing researchers to create a record of their various scholarly contributions in a single place. Please note that ORCID IDs are now *required* for all authors. At resubmission of your final files, please be sure to provide your ORCID ID and those of all co-authors.

Please note that JCB now requires authors to submit Source Data used to generate figures containing gels and Western blots with all revised manuscripts. This Source Data consists of fully uncropped and unprocessed images for each gel/blot displayed in the main and supplemental figures. For assays performed using capillary electrophoresis and/or immunoassay-based detection, authors should instead provide the electropherogram graph(s) for each experiment, plotting fluorescence/chemiluminescence intensity vs. molecular weight/size. Please be sure to provide one Source Data file for each figure gels, blots, and/or capillary electrophoresis assays along with your revised manuscript files. File names for Source Data figures should be alphanumeric without any spaces or special characters (i.e., SourceDataF#, where F# refers to the associated main figure number or SourceDataFS# for those associated with Supplementary figures). For traditional gels and blots, the lanes of the gels/blots should be labeled as they are in the associated figure, the place where cropping was applied should be marked (with a box), and molecular weight/size standards should be labeled wherever possible. For capillary electrophoresis assays, each trace in the graph should be color-coded and labeled to indicate which protein, gene, or sample is being measured (please try to avoid red/green combinations to accommodate our color-blind readers).

Journal of Cell Biology now requires a data availability statement for all research article submissions. These statements will be published in the article directly above the Acknowledgments. The statement should address all data underlying the research presented in the manuscript. Please visit the JCB instructions for authors for guidelines and examples of statements at (<https://rupress.org/jcb/pages/editorial-policies#data-availability-statement>).

B. FINAL FILES:

****It is JCB policy that if requested, original data images must be made available to the editors. Failure to provide original images upon request will result in unavoidable delays in publication. Please ensure that you have access to all original data images prior to final submission.****

****The license to publish form must be signed before your manuscript can be sent to production. A link to the electronic license to publish form will be sent to the corresponding author only. Please take a moment to check your funder requirements before choosing the appropriate license.****

Thank you for your attention to these final processing requirements. Please revise and format the manuscript and upload materials within 7 days. If you need an extension for whatever reason, please let us know and we can work with you to determine a suitable revision period.

Thank you for this interesting contribution, we look forward to publishing your paper in Journal of Cell Biology.

Sincerely,

Aaron Straight
Monitoring Editor

Andrea L. Marat
Deputy Editor

Journal of Cell Biology

Reviewer #1 (Comments to the Authors (Required)):

The authors have significantly expanded the manuscript with several additional experiments, and the experiments reported in the initial submission are now better controlled and more clearly explained. The authors have addressed all of my concerns and added several interesting new insights that make this study interesting and novel. I am happy to support publication.

Reviewer #2 (Comments to the Authors (Required)):

Overall, the revised manuscript provides a much more complete story that advances our understanding of how centromeres respond to cellular quiescence. The manuscript demonstrates that loss of centromere proteins through the cessation of the transcriptional programs during quiescence, and define how centromere re-assembly progresses following re-entry into the cell cycle. New data highlights the unique regulation of CENP-C. Very similar findings showing distinct regulation of CENP-C from other CCAN proteins is previously published in Khurana et al. (MCB 2024) and should be referenced. I support publication of the manuscript.